# Enhancer hijacking activates oncogenic transcription factor NR4A3 in acinic cell carcinomas of the salivary glands

Florian Haller [1], Matthias Bieg[2,3], Rainer Will[4], Cindy Körner[5], Dieter Weichenhan[6], Alexander Bott[5], Naveed Ishaque[2,3], Pavlo Lutsik [6], Evgeny A. Moskalev[1], Sarina K. Mueller[7], Marion Bähr[6], Angelika Woerner[5], Birgit Kaiser[4], Claudia Scherl[7], Marlen Haderlein[8], Kortine Kleinheinz[9], Rainer Fietkau[8], Heinrich Iro[7], Roland Eils[2,3,10,11], Arndt Hartmann[1], Christoph Plass[6], Stefan Wiemann[4,5] & Abbas Agaimy[1]

The molecular pathogenesis of salivary gland acinic cell carcinoma (AciCC) is poorly understood. The secretory Ca-binding phosphoprotein (SCPP) gene cluster at 4q13 encodes structurally related phosphoproteins of which some are specifically expressed at high levels in the salivary glands and constitute major components of saliva. Here we report on recurrent rearrangements [t(4;9)(q13;q31)] in AciCC that translocate active enhancer regions from the SCPP gene cluster to the region upstream of *Nuclear Receptor Subfamily 4 Group A Member 3 (NR4A3)* at 9q31. We show that NR4A3 is specifically upregulated in AciCCs, and that active chromatin regions and gene expression signatures in AciCCs are highly correlated with the NR4A3 transcription factor binding motif. Overexpression of NR4A3 in mouse salivary gland cells increases expression of known NR4A3 target genes and has a stimulatory functional effect on cell proliferation. We conclude that NR4A3 is upregulated through enhancer hijacking and has important oncogenic functions in AciCC.

[1] Institute of Pathology, University Hospital Erlangen, Friedrich-Alexander University Erlangen-Nuremberg, Krankenhausstr. 8-10, 91054 Erlangen, Germany. [2] Center for Digital Health, Berlin Institute of Health and Charité – Universitätsmedizin Berlin, Kapelle-Ufer 2, 10117 Berlin, Germany. [3] Heidelberg Center for Personalized Oncology (DKFZ-HIPO), Im Neuenheimer Feld 280, 69120 Heidelberg, Germany. [4] Genomics and Proteomics Core Facility, German Cancer Research Center (DKFZ), Im Neuenheimer Feld 580, 69120 Heidelberg, Germany. [5] Division of Molecular Genome Analysis, German Cancer Research Center (DKFZ), Im Neuenheimer Feld 580, 69120 Heidelberg, Germany. [6] Division of Cancer Epigenomics, German Cancer Research Center (DKFZ), Im Neuenheimer Feld 280, 69120 Heidelberg, Germany. [7] Department of Otorhinolaryngology, Head & Neck Surgery, University Hospital Erlangen, Friedrich-Alexander University Erlangen-Nuremberg, Waldstrasse 1, 91054 Erlangen, Germany. [8] Department of Radiation Therapy, University Hospital Erlangen, Friedrich-Alexander University Erlangen-Nuremberg, Universitätsstrasse 27, 91054 Erlangen, Germany. [9] Division of Theoretical Bioinformatics (B080), German Cancer Research Center (DKFZ), Im Neuenheimer Feld 280, 69120 Heidelberg, Germany. [10] Health Data Science Unit, University Hospital Heidelberg, Im Neuenheimer Feld 267, 69120 Heidelberg, Germany. [11] Translational Lung Research Center Heidelberg, German Center for Lung Research, University of Heidelberg, Im Neuenheimer Feld 156, 69120 Heidelberg, Germany. These authors contributed equally: Florian Haller, Matthias Bieg, Stefan Wiemann, Abbas Agaimy. Correspondence and requests for materials should be addressed to F.H. (email: florian.haller@uk-erlangen.de) or to S.W. (email: s.wiemann@dkfz-heidelberg.de)

A group of structurally related genes encoding secretory Ca-binding phosphoproteins (SCPPs) map to a genomic region of ~ 750 kb on chromosome 4q13, i.e., the SCPP gene cluster[1]. These genes evolved from a common ancestor gene by tandem duplication events[1]. The encoded salivary proteins, milk caseins and enamel matrix proteins are functionally related through their ability to regulate the calcium phosphate levels in the extracellular environment. The function of the salivary proteins includes protection of tooth enamel by stabilizing saliva supersaturated with calcium salts, as well as antimicrobial and antifungal properties[1]. The expression of the majority of the salivary gland genes within the SCPP gene cluster (e.g., *STATH*, *HTN1*, *HTN3*, *ODAM*, *FDCSP*, and *MUC7*) is mostly restricted to the salivary glands, where they rank among the most highly expressed genes (https://gtexportal.org/home/).

Acinic cell carcinoma (AciCC) shows serous differentiation closely resembling normal acini of salivary glands, which produce major components of saliva[2]. While no recurrent molecular genetic event in AciCC has been reported yet[2], several other salivary gland neoplasms are characterized by specific recurrent genomic rearrangements[3,4]. These typically result in the generation of oncogenic chimeric fusion genes, e.g., the *NFIB-MYB* gene fusion in adenoid cystic carcinomas (ACC)[5]. Only recently, an alternate mechanism of oncogenic upregulation of the same transcription factor MYB through genomic rearrangements involving the translocation of enhancer regions has been demonstrated in ACC lacking the *NFIB-MYB* gene fusion[6]. Similarly, "regulatory rearrangements" driving oncogene overexpression through enhancer hijacking or insulator dysfunction are increasingly recognized in other cancer types[7,8].

For the current study, we employ comprehensive genomic, transcriptomic, and epigenomic profiling of AciCCs and normal parotid gland tissue to decipher the molecular genetic events in AciCC. We show that AciCCs harbor recurrent and specific rearrangements [t(4;9)(q13;q31)] that bring active enhancer regions from the SCPP gene cluster in close proximity to the transcription start site (TSS) of transcription factor *Nuclear Receptor Subfamily 4 Group A Member 3 (NR4A3)* at 9q31. The genomic breakpoints within the SCPP gene cluster correlate with high H3K27ac active chromatin marks that are present in both normal parotid gland and AciCC tumor tissues, and we demonstrate significant enhancer activity within the breakpoint region in a functional analysis. We show that NR4A3 is specifically and consistently upregulated in AciCCs. There is an increase in active chromatin marks and upregulation of gene expression in AciCC tumor tissues compared to normal parotid gland that is significantly correlated with the NR4A3 transcription factor binding motif. Functional analyses of mouse salivary gland and human mammary gland cells with NR4A3 overexpression show significant effects on gene expression including known NR4A3 target genes, as well as stimulatory effects on cell proliferation and cell cycle. We conclude that the genomic rearrangements [t(4;9)(q13;q31)] are the initiating tumorigenic events in AciCCs and that these rearrangements enable upregulation of NR4A3 through inappropriate influence of active chromatin regions derived from the SCPP gene cluster. We suggest that NR4A3 upregulation is an early oncogenic event in AciCCs through constitutive stimulation of cell proliferation.

## Results

### Identification of recurrent rearrangements [t(4;9)(q13;q31)].
Whole Genome Sequencing (WGS) of four paired tumor/normal AciCC cases and tumor WGS in two additional AciCCs revealed that copy number changes were few and broadly distributed among the tumor genomes without recurrent gains or losses present in all

cases (Fig. 1a, Supplementary Data 1). The tumors carried very few somatic non-synonymous variants in the exonic fraction (SNVs: mean 14, range 7–17; InDels mean: 3, range 2–7), and no gene was mutated in more than one tumor (Supplementary Table 1). In contrast, a specific inter-chromosomal rearrangement [t(4;9)(q13; q31)] was present in all six AciCCs (Fig. 1b). The 4q13 breakpoints were distributed within ~ 340 kb at the SCPP gene cluster spanning eight genes (Fig. 1c, upper panel left side). All of the 9q31 breakpoints were located upstream of *NR4A3*, with four events located within a ~ 600 bp region directly upstream of the TSS (Fig. 1c, upper panel right side). We thus focused on the rearrangement [t (4;9)(q13;q31)] and *NR4A3* as a potential driver in AciCC, and established a hybrid capture based next generation sequencing approach with probes covering the ~ 315 kb upstream of the *NR4A3* TSS to identify nine additional AciCCs with a [t(4;9)(q13;q31)] translocation (Fig. 1c, middle panel).

**Transcription factor NR4A3 is upregulated in AciCCs**. We further employed Fluorescence in-Situ Hybridization (FISH) using a break-apart probe flanking the *NR4A3* gene locus (Fig. 1c lower panel right side, Supplementary Figure 1) to evaluate the frequency of this translocation event in a total of 29 AciCCs and 75 other salivary gland neoplasms. Split signals indicating a *NR4A3* gene locus rearrangement were present in 24 of 28 evaluable AciCCs (86%), while no *NR4A3* translocation was observed in any of 75 non-AciCC salivary gland tumors (Fig. 1d, Supplementary Table 2). Whole transcriptome RNA-sequencing analysis of ten AciCCs (Supplementary Data 2) revealed no coding chimeric fusion transcripts as described previously[9]. However, as compared to normal parotid salivary gland tissue, significant upregulation of *NR4A3*, the gene closest to the 9q31 breakpoints, was determined in all AciCCs, while there was no significant difference in the expression of other genes located upstream and further downstream of the 9q31 breakpoints (Fig. 1e). Twenty-eight of 29 AciCCs (97%) showed nuclear immunostaining for NR4A3 (Supplementary Table 2), while NR4A3 was not detectable in normal salivary gland tissue (Fig. 1f).

**The SCPP gene cluster harbors highly active chromatin marks**. The lack of a coding gene fusion and the observation that the 4q13 breakpoints were rearranged in either forward or reverse orientation to the *NR4A3* upstream region directed us towards regulatory chromatin marks (Fig. 2, Supplementary Data 3), which function independent of the genomic sequence orientation. We employed Chromatin Immunoprecipitation followed by sequencing (ChIP-seq) using antibodies against core histone modifications H3K27ac (active promoter/enhancer), H3K4me3 (active promoter), H3K27me3 (repressed chromatin), and against the transcription factors NR4A3 and CCCTC-Binding Factor (CTCF), as well as whole genome bisulfite sequencing (Supplementary Data 4), to decipher epigenetic regulatory marks around the breakpoint regions. This data were complemented with published data on topologically associated domains (TADs) and chromatin contacts (HiC)[10]. We observed enrichment of active H3K27ac and H3K4me3 chromatin marks and low DNA methylation levels broadly distributed throughout the 4q13 SCPP gene cluster in normal parotid gland tissue, consistent with abundant expression of the SCPP salivary gland genes mapping there (Fig. 2 left panel). Comparison with published chromatin contact data[10] indicated three sub-TAD domains within a larger TAD, encompassing the majority of the highly expressed salivary gland genes (e.g., *STATH*, *HTN3*, *HTN1*) within a single sub-TAD region characterized by abundant active chromatin marks, with inversely oriented CTCF binding sites correlating well with the borders of the sub-TAD regions. In contrast, there were both

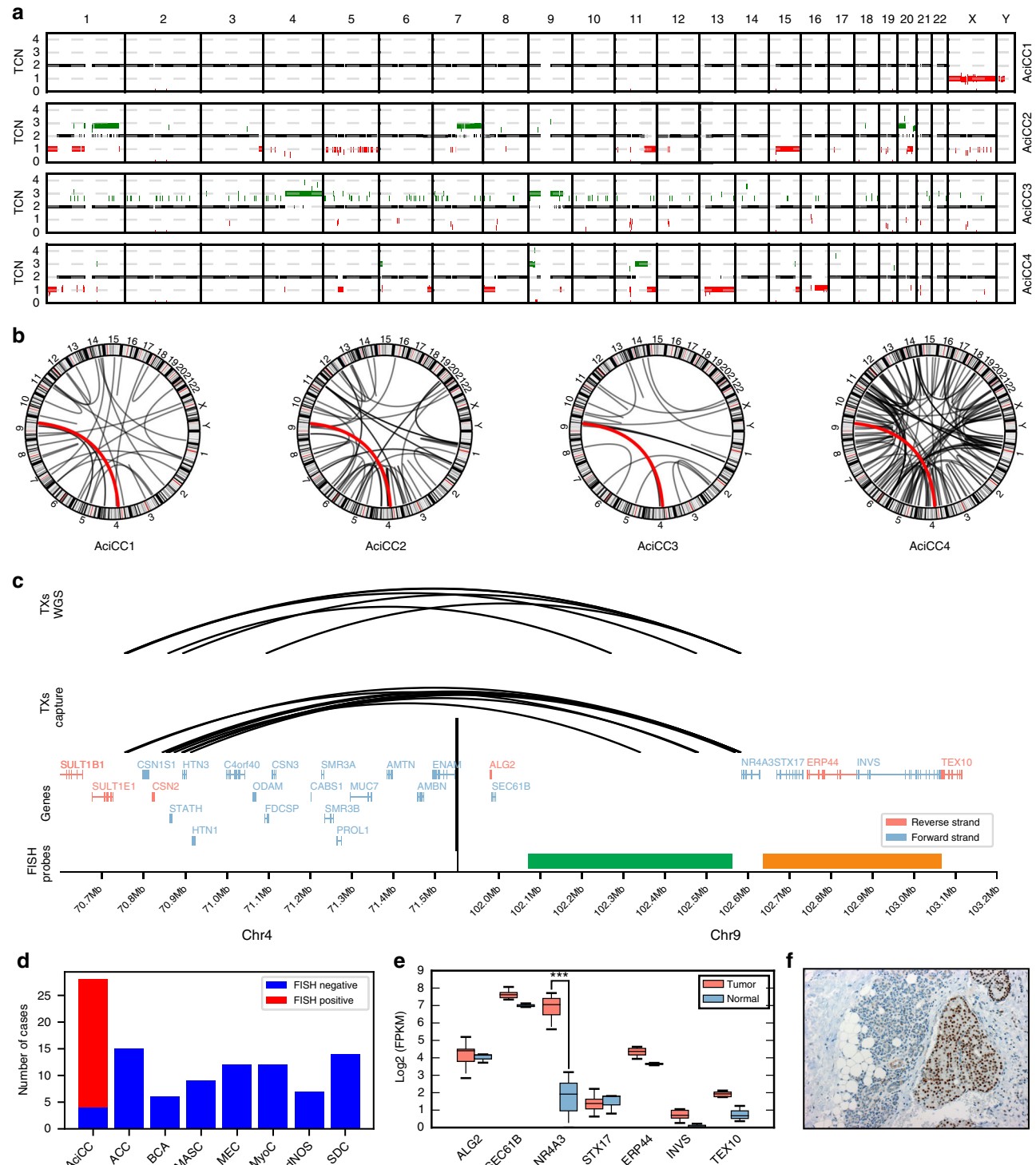

active H3K27ac/H3K4me3 and repressive H3K27me3 chromatin marks at the 9q31 locus upstream of the *NR4A3* TSS and very-low DNA methylation levels at the promoter region in the normal parotid tissue, indicating a bivalent/poised promoter (Fig. 2 right panel). Comparing three AciCC samples with normal parotid tissue, the chromatin marks, as well as DNA methylation patterns were overall well-preserved in the tumors, with slightly lower repressive H3K27me3 chromatin marks at the *NR4A3* promoter region and lower DNA methylation levels at the *NR4A3* gene body in two samples. ChIP-seq for NR4A3 binding sites revealed several peaks at the SCPP gene cluster in the tumor tissues that

corresponded well with active chromatin marks, while there was no significant NR4A3 binding signal in the normal parotid tissue. Regarding the *NR4A3* gene locus, there were NR4A3 binding peaks at the upstream region of the *NR4A3* gene locus both in the normal parotid and tumor tissues.

**Three distinct patterns of rearrangements [t(4;9)(q13;q31)].** Notably, in all six AciCCs analyzed by WGS, the 4q13 genomic breakpoints mapped to regions with an enrichment of active chromatin marks associated with abundantly expressed salivary

**Fig. 1** Identification of recurrent rearrangements t(4;9)(q13;q31) in AciCCs. **a** Plots of copy number changes in four AciCCs with paired tumor-normal whole genome sequencing (WGS) show gains (green) and losses (red) in the individual tumors. **b** Circos plots of translocations in the four cases, with recurrent rearrangements t(4;9)(q13;q31) highlighted in red. **c** Upper panel: Detailed mapping of t(4;9)(q13;q31) chromosomal breakpoints of the four AciCCs with paired tumor-normal WGS and two AciCCs with only tumor WGS demonstrate the distribution of 4q13 breakpoints among ~ 340 kbps spanning eight different genes at the SCPP gene cluster (left side), and clustering of the 9q31 breakpoints upstream of the NR4A3 gene locus (right side). Middle panel: Detailed mapping of t(4;9)(q13;q31) chromosomal breakpoints of nine additional AciCCs with tumor hybrid capture sequencing data confirms the pattern of 4q13 breakpoints within the SCPP gene cluster (left) and upstream of the NR4A3 gene locus (right). Lower panel: SCPP gene cluster (left side), NR4A3 and neighboring genes (right side) with green and orange bars indicating the location of NR4A3 break apart FISH probes. **d** Absolute number of cases with genomic rearrangements of the NR4A3 gene locus in 28 AciCCs and 75 other salivary gland neoplasms analyzed by FISH. **e** mRNA expression (log2 FPKM values) of NR4A3 and neighboring genes in ten tumor samples and three normal parotid gland samples, with only NR4A3 showing a significant upregulation (deSeq2[47]; ***$P < 0.001$; Box-plot center line: median; bounds of box: 25 and 75% quantiles; whiskers: minimum and maximum values). **f** NR4A3 immunohistochemistry demonstrating strong nuclear expression in AciCC tumor tissue (right), but absence in normal parotid gland tissue (left). Source data for Figs. 1b, 1c and 1e are provided as a Source Data file. Source Data for Fig. 1d are provided in Supplementary Table 2. TCN, Tumor Copy Number; TXs, Translocations; SCPP, secretory Ca-binding phosphoprotein; AciCC, Acinar cell carcinoma; ACC, adenoid cystic carcinoma; BCA, basal cell adenocarcinoma; MASC, mamma-analog secretory carcinoma; MyoC, myoepithelial carcinoma; AdNOS, adenocarcinoma not otherwise specified; SDC, salivary duct carcinoma

gland genes (Fig. 2 left panel). Adding the additional nine AciCCs with rearrangement [t(4;9)(q13;q31)] identified by the hybrid capture assay, three distinct patterns of rearrangements could be established (Fig. 3). In four AciCCs (AciCC5, AciCC4, AciCC3, AciCC19), the 4q13 breakpoint was located in a small genomic region that fell between two inversely oriented CTCF binding sites that were characterized by active H3K27ac peaks and correlated with the border between two sub-TAD regions as indicated by published chromatin contact data (Figs. 2 and 3). In all four samples the breakpoints were orientated in the same direction, e.g., the 4q13 reverse strand was fused to the 9q31 forward strand, so that in all four cases the resulting rearranged sequence included the genomic region of the SCPP gene cluster. The second and most frequent pattern was present in ten AciCCs, which all harbored the 4q13 breakpoint within a genomic region encompassing the three most abundantly expressed salivary gland genes STATH, HTN3, and HTN1 with correlating highly active chromatin marks (Fig. 3). In contrast to the tumors with the first pattern, both orientations of rearrangements of the 4q13 genomic region were observed in this group, with three tumors having the 4q13 forward strand fused to the NR4A3 upstream region, and seven the 4q13 reverse strand. The third pattern was identified in only one sample (AciCC2) where the 4q13 breakpoint fell directly within the first intron of the salivary gland gene FDCSP, but no chimeric fusion reads between FDCSP and NR4A3 were detected on the RNA level. Further exploiting the ChIP-seq data, we next modeled the potential interaction of active H3K27ac and H3K4me3 chromatin marks from the 4q13 SCPP gene cluster and the NR4A3 gene locus as a result of the genomic rearrangements for three individual tumor samples AciCC3, AciCC1 and AciCC2, representing the three different breakpoint patterns, respectively, (Fig. 4). In AciCC3 (Fig. 4b) representing the second most common group with breakpoints between two CTCF binding sites, active chromatin marks from the SCPP gene cluster were situated directly at the breakpoint as well as ~ 90 kb, ~ 130 kb and ~ 290 kb upstream of the breakpoint, with the latter three peaks ranking among the highest 5% of peaks and thus fulfilling the criteria of super-enhancers (Supplemental Figure S5a). In AciCC1 (Fig. 4c) representing the most frequently observed pattern with breakpoints within the genomic region encoding for STATH, HTN3 and HTN1, active chromatin marks derived from the 4q13 SCPP gene cluster were situated directly at the breakpoint, as well as ~ 25 kb upstream of the breakpoint, and these chromatin marks ranked high within the highly active chromatin marks in this sample but did not fulfill the criteria of super-enhancers (Supplementary Figure 2). In AciCC2 (Fig. 4d) representing its own pattern with the breakpoint directly within the FDCSP gene,

active chromatin marks from the FDCSP promoter region were directly located at the breakpoint region, as well as more distal at ~ 28 kb, ~ 174 kb, and ~ 228 kb upstream of the breakpoint. In contrast, no active chromatin marks were found in normal parotid tissue without genomic rearrangement within the same distance upstream of the NR4A3 TSS (Fig. 4a). In all three tumors analyzed, the active chromatin marks derived from the SCPP gene cluster were partially accompanied by NR4A3 peaks, and there was a reduction of inhibitory H3K27me3 peaks at the NR4A3 promoter region. Additionally, a CTCF binding site 13.5 kb upstream of the NR4A3 gene locus was separated from the ultimate NR4A3 promoter region in all three AciCCs, possibly disrupting an insulator function.

**Analysis of enhancer activity within the SCPP gene cluster.** Since no minimal overlapping genomic region within the SCPP gene cluster could be defined among the 15 AciCCs with recurrent [t(4;9)(q13;q31)], we focused on the most common group with breakpoints within the genomic region encoding for STATH, HTN3, and HTN1. Putative enhancer regions were identified by presence of active chromatin marks within this region and comparison with a published data set[11]. We focused on two proximal and distal regions upstream of the STATH TSS with highly active chromatin marks in our normal parotid gland ChIP-seq data, including a putative enhancer region predicted by published chromatin segmentation data within the distal region (Fig. 5). Ten fragments with a size of 340–846 bp mapping within these regions were cloned and analyzed for enhancer activity using a dual luciferase reporter assay. Indeed, the fragment mapping to genomic coordinates chr4:70849125-70849971 correlated to the predicted enhancer region[11] and displayed significant enhancer activity in this analysis (two-sided T-Test, $P <$ 0.01, Fig. 5).

**Gene expression correlates with NR4A3 binding motif.** To analyze the correlation between NR4A3 transcription factor upregulation and global mRNA expression in AciCCs, we compared whole transcriptome RNA sequencing and ChIP-seq data. Gene expression analysis of ten AciCCs and three normal parotid tissues identified 827 and 404 genes that were significantly up- and downregulated in AciCCs vs. normal parotid tissues, respectively (Fig. 6a, Supplementary Data 2). Functional enrichment analysis of genes that were upregulated in the tumors revealed a significant enrichment of genes involved in DNA replication, cell division, and DNA damage response (annotation cluster 1 to 4 and 6), and of genes involved in inflammatory response and complement

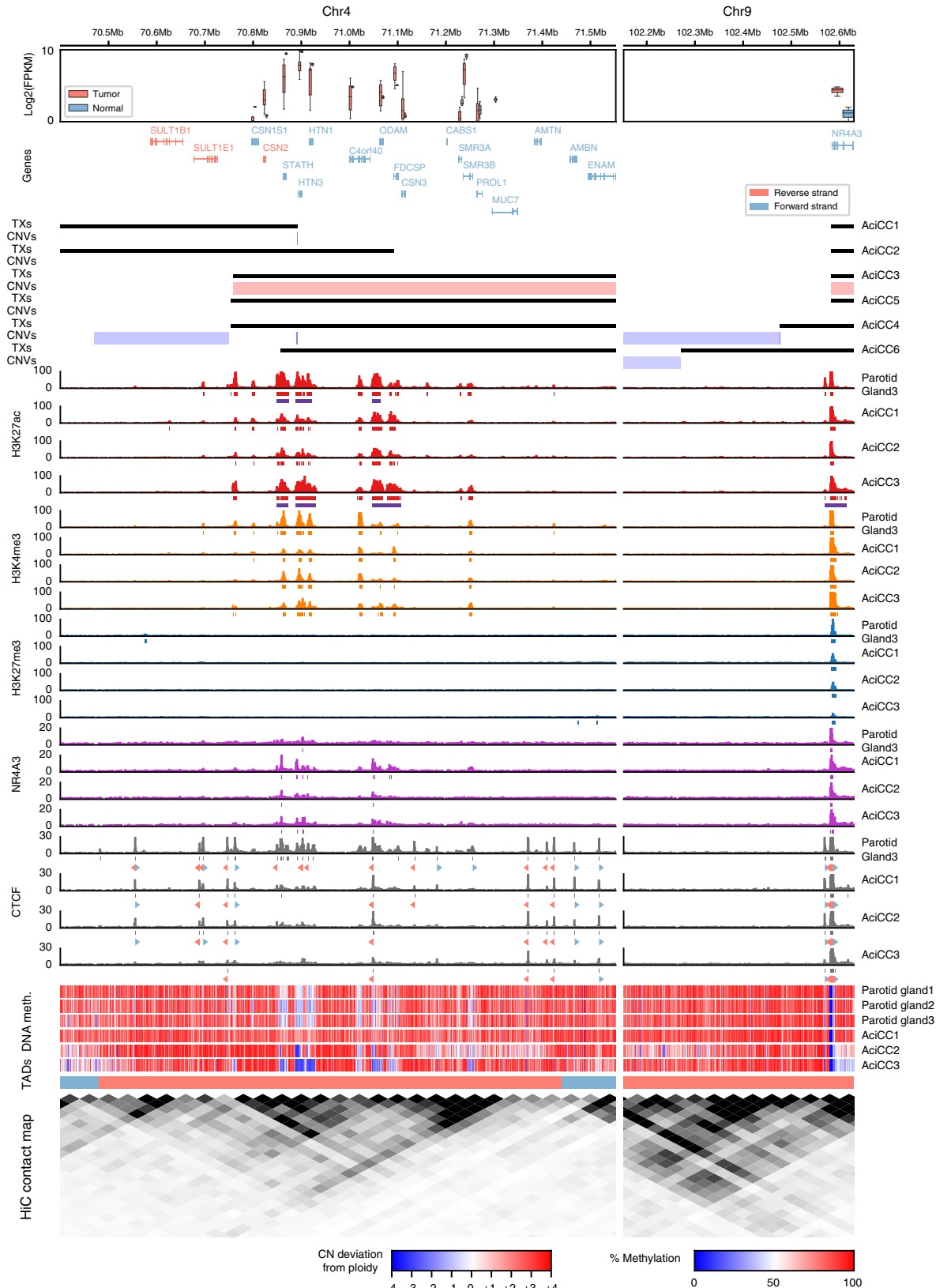

activation (annotation cluster 5 and 7). Among the downregulated genes, there was a significant enrichment of genes involved in GTPase activation and salivary secretion (annotation cluster 1 and 2, respectively). A comparison of differentially expressed genes with differential ChIP-seq peaks for active histone mark H3K27ac and NR4A3 transcription factor revealed a positive correlation between differential H3K27ac and NR4A3 peaks (Supplementary Figure 3), and a significant enrichment of higher H3K27ac and NR4A3 peaks was observed among upregulated genes in the tumors. Approximately 25% and 15% of upregulated genes were associated with upregulated H3K27ac and NR4A3 peaks in the tumors, respectively (Fig. 6b). While ~ 25% of downregulated genes were associated with

**Fig. 2** Genomic breakpoints in AciCCs correlate with active chromatin marks and NR4A3 binding sites at the 4q13 SCPP gene cluster. Summary of (top to bottom) mRNA expression (log2(FPKM)), gene loci (Genes), genomic translocation breakpoints (TXs), copy-number variations (CNVs), active (H3K27ac, H3K4me3) and repressive (H3K27me3) histone marks, NR4A3 binding sites, CTCF binding sites, DNA methylation, topologically associated domains (TADs), and chromatin contacts (HiC)[10] for the chromosomal regions surrounding the 4q13 (left panel) and 9q31 (right panel) breakpoints in normal parotid gland and AciCC tumor tissues. mRNA expression is shown for ten AciCC tumors and three normal parotid gland samples in red and blue, respectively (Box-plot centre line: median; bounds of box: 25 and 75% quantiles; whiskers: extend to last value greater than Q1–1.5*IQR, and last value less than Q3+1.5*IQR respectively. Here IQR is the inter quartile range, Q1 is the first, and Q3 the third quartile). Black bars demonstrating the translocations correspond to genomic material included in the rearrangement, with red and blue bars indicating gains and losses, respectively. For each ChIP-seq experiment, ChIP signals (barcharts), as well as corresponding peaks (directly below ChIP signals) are shown. For H3K27ac, the super-enhancer peaks are shown in addition (purple). Furthermore, CTCF motifs within CTCF peaks are shown (below CTCF peaks). Blue arrows indicate motifs on the forward strand, whereas red arrows indicate motifs on the reverse strand. The DNA methylation tracks show the average methylation within 1 kb binned regions. Different TADs are indicated in different colors

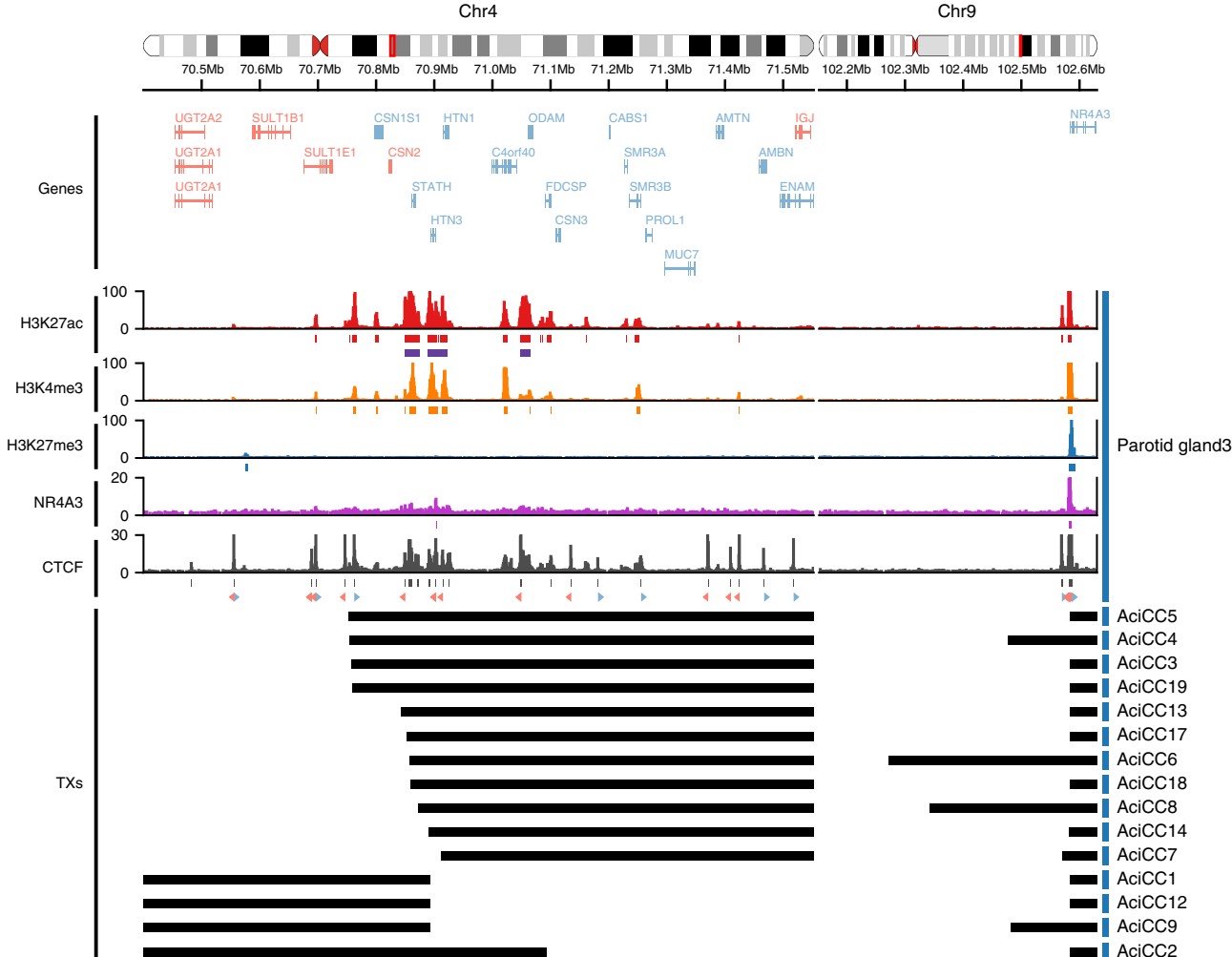

**Fig. 3** Detailed characterization of rearrangements t(4;9)(q13;q31) in 15 AciCCs identify three different breakpoint patterns. Detailed presentation of rearrangements in 15 AciCCs with t(4;9)(q13;q31), with black bars indicating genomic material included in the rearrangement. Active (H3K27ac, H3K4me3) and repressive (H3K27me3) histone marks, NR4A3 binding sites and CTCF binding sites in normal parotid gland tissue are presented for comparison. Source data for translocation breakpoints are provided as a Source Data file

reduced H3K27ac peaks, there were only few reduced NR4A3 peaks in the tumor tissues, and only 2% of the downregulated genes were associated with reduced NR4A3 peaks. Motif analysis of differential H3K27ac peaks in the three tumors vs. the normal parotid tissue identified a significant enrichment of the NGFI-B response element (NBRE) (Fig. 6b), which is the main binding motif for NR4A family members[12]. Correspondingly, de-novo motif analysis of enriched NR4A3 peaks in ChIP-seq data from three AciCC tumor samples revealed the same motif in all three cases (Fig. 6b). The known

NR4A3 target genes *CCND1*[13] and *ENO3*[14] were higher expressed in AciCC tissues on the mRNA level compared to normal parotid gland tissue (Supplementary Figure 4), and this was confirmed on the protein level by immunohistochemistry (Supplementary Figure 5, Supplementary Table 2).

**Functional effects of NR4A3 overexpression in vitro.** To demonstrate the driver activity of NR4A3 in cellular systems, we stably transduced immortalized primary mouse submandibular

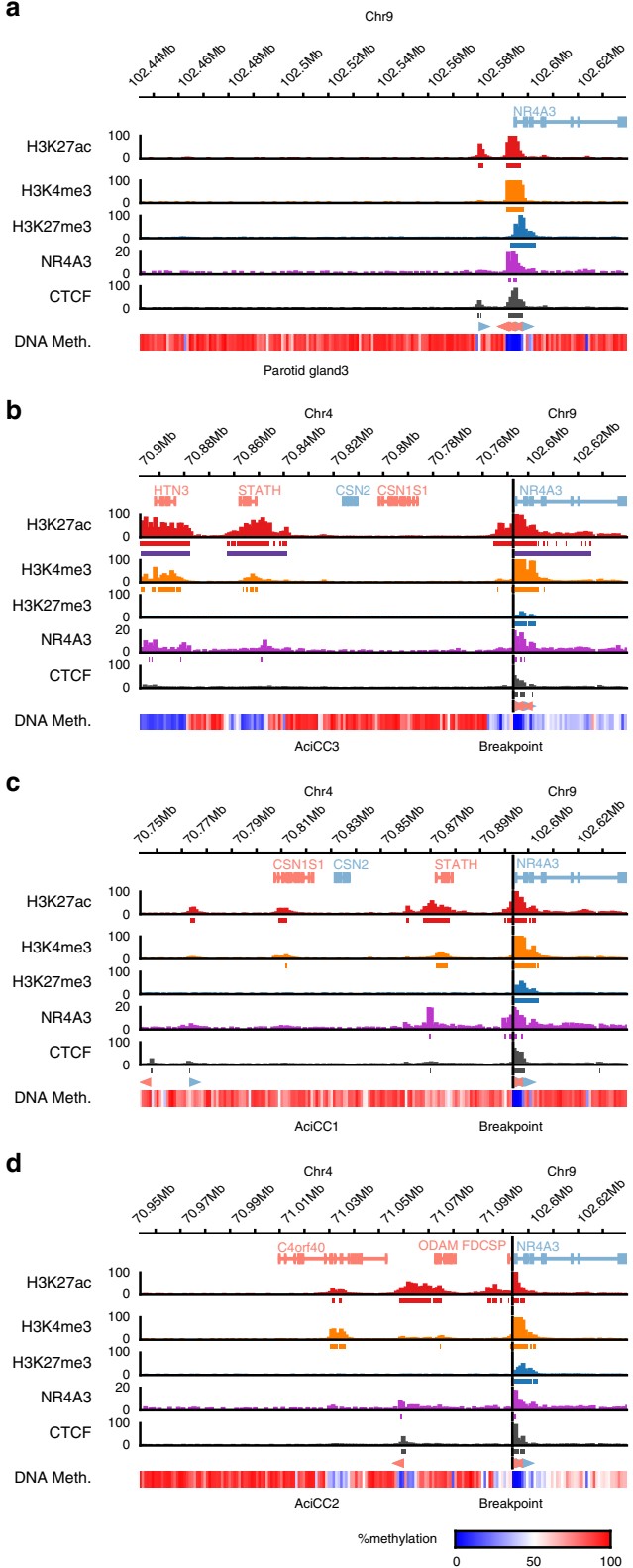

**Fig. 4** Rearrangements t(4;9)(q13;q31) in AciCCs juxtapose the *NR4A3* gene locus proximal to active enhancers and NR4A3 binding sites. Detailed presentation of active (H3K27ac, H3K4me3) and repressive (H3K27me3) histone marks, NR4A3 binding sites, CTCF binding sites, and DNA methylation levels in normal parotid gland (**a**) and three distinct AciCC tumor samples AciCC3 (**b**), AciCC1 (**c**) and AciCC2 (**d**). For each ChIP-seq experiment, ChIP signals (barcharts) as well as corresponding peaks (directly below ChIP signals) are shown. For H3K27ac, the super-enhancer peaks are shown in addition (purple). Furthermore, CTCF motifs within CTCF peaks are shown (below CTCF peaks). Blue arrows indicate motifs on the forward strand, whereas red arrows indicate motifs on the reverse strand. The DNA methylation tracks show the average methylation within 1 kb binned regions

NR4A3 was found expressed only in the *NR4A3* stably transduced cells both at mRNA and protein levels but not in the controls. Mouse Nr4a3 was not detected in any condition. Mouse Ccnd1 and Eno3 mRNAs were found significantly upregulated and mouse Ccnd1 and Eno3 proteins were present at higher levels specifically in NR4A3 overexpression conditions (Fig. 6d). This was validated by qRT-PCR for Ccnd1 and Eno3 (Supplementary Figure 8), and by western blot for Ccnd1 (Supplementary Figure 9). Furthermore, there was a significant upregulation of 432 genes and downregulation of 561 genes in NR4A3 expressing cells compared to the control cells on the mRNA level (source data file). Nine proteins, including NR4A3, were found expressed only in the cells stably transduced with NR4A3, while 17 proteins were exclusively detected in the proteomic datasets of controls (Supplementary Table 3). Second, to test for potential functional consequences of NR4A3 expression in the mouse salivary gland cells, cell viability and proliferation were evaluated. Indeed, expression of NR4A3 increased the number of cells accumulating with time and also enhanced the fraction of cells in S-phase of the cell cycle (Fig. 6e). To verify that these observations are not cell line or species specific, we stably transduced the human mammary gland cell line MCF10A with the *NR4A3* coding sequence and then selected clones with and without *NR4A3* expression (qRT-PCR) for functional testing. Consistent with the data obtained in the mouse salivary gland cell line model, the MCF10A cells proliferated significantly more when NR4A3 was expressed and a similar increase in the fraction of cells in S-phase was observed (Fig. 6f).

## Discussion

NR4A3 is a member of the NR4A subfamily of nuclear receptors that are involved in manifold physiological processes including cell proliferation, differentiation, and metabolism[10]. Through binding to the NBRE motif[10], NR4A proteins are transcription factors that directly modulate gene expression, but can also be part of larger trans-repressive complexes[15,16]. A fusion gene comprised of *NR4A3* and *EWSR1* results from a recurrent translocation [t(9;22)(q31;q12)] in extraskeletal myxoid chondrosarcoma and is the oncogenic driver event in this entity[17]. In contrast, NR4A3 has been described as tumor suppressor in hematopoietic neoplasms, e.g., acute myeloic leukemia[18]. Likely, the cellular context and the presence or absence of further co-factors determines the effect of NR4A3 function on gene transcription and phenotypes. Only few genes including the cell cycle regulator *Cyclin D1* (*CCND1*)[13], the metabolic enzyme *Enolase 3* (*ENO3*)[14], the secreted serum and extracellular matrix glycoprotein *Vitronectin* (*VTN*)[19], and the nuclear receptor and transcriptional regulator *Peroxisome Proliferator Activated Receptor Gamma* (*PPARG*)[20] have been functionally validated as direct targets of NR4A3 or the NR4A3-EWSR1 fusion protein in

gland (SG) cells either with the human *NR4A3* open reading frame (ORF) or with red firefly luciferase coding regions and investigated molecular and phenotypic effects. First, we performed RNA-seq and whole mass spectrometry proteome analysis for unbiased determination of mRNA expression (Supplementary Figure 6, Supplementary Data 5) and protein abundance (Supplementary Figure 7), respectively. Human

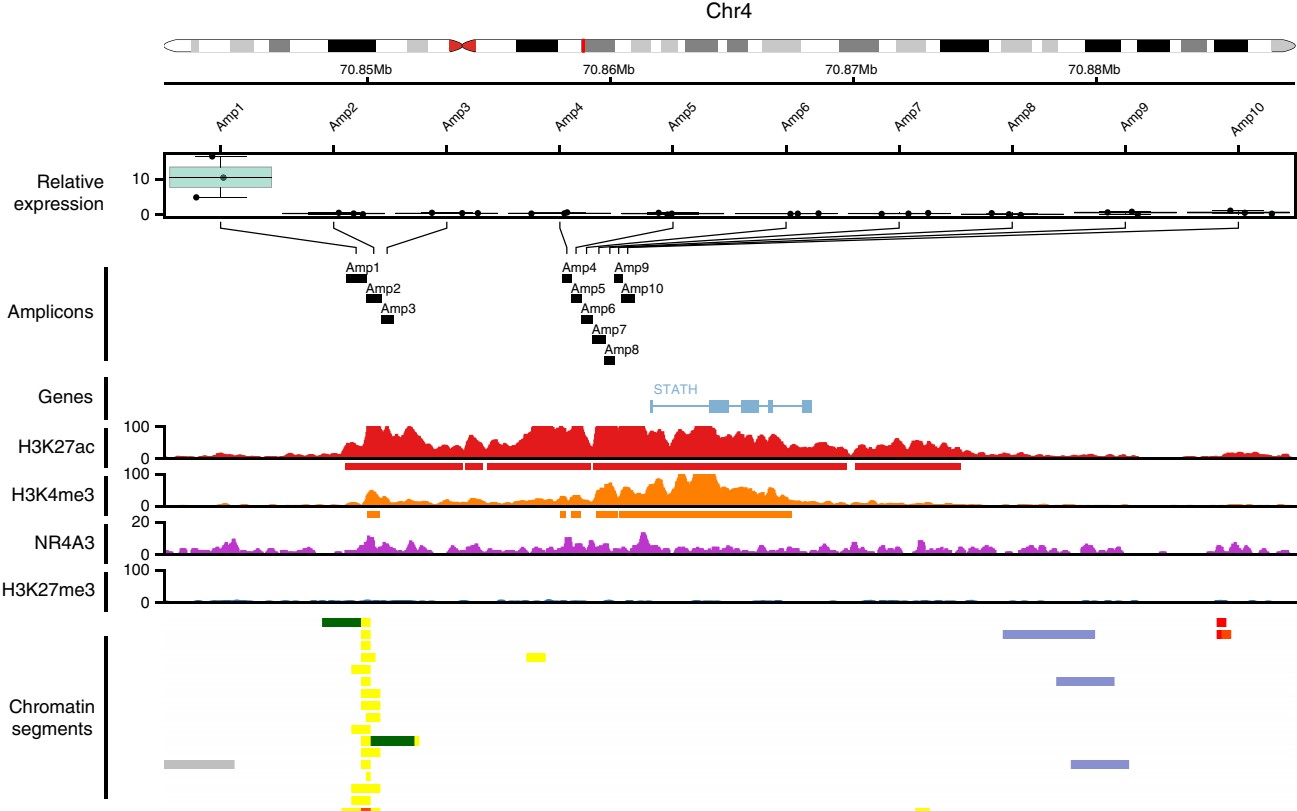

**Fig. 5** Analysis of enhancer activity in a genomic region shared by AciCCs with t(4;9)(q13;q31) rearrangement. Correlation of enhancer activity determined in the dual luciferase reporter assay for ten different genomic fragments (upper panel) with chromatin marks determined by ChIP-seq analyzes in parotid salivary gland tissue (middle panel) and putative enhancer regions based on genome wide chromatin segmentation (lower panel)[(11)]. For each fragment, relative expression values for three replicates normalized to the control vector are presented. Box-plot center line: median; bounds of box: 25 and 75% quantiles; whiskers: minimum and maximum values). One-tailed Student's *t*-test; Amplicon 1: *P* < 0.001. Source data are provided as a Source Data file

different in vitro cell models. For the current study, we focused on CCND1 and ENO3 and observed consistently higher mRNA expression and protein levels in human AciCC tissues. Immortalized mouse salivary gland cells stably transduced with human NR4A3 revealed higher mRNA expression and protein levels of mouse Ccnd1 and Eno3 compared to controls, thus confirming a transcriptional effect of NR4A3 upregulation on its known target genes in this cell model. Upregulation of the cell cycle regulator CCND1 is well in line with published data showing that NR4A3 is a potent activator of vascular smooth muscle cell and hepatocyte proliferation[21–23], with Cyclin D1 representing a direct transcriptional target of NR4A3[13,22,23]. This also correlates with an earlier study reporting on the expression of Cyclin D1 in AciCC[24]. Employing two different cell models derived from mouse salivary gland cells and human mammary gland cells, we observed a stimulatory functional effect of NR4A3 overexpression on cell proliferation and the cell cycle in both models, further supporting its potential oncogenic role. However, we also observed a strong impact of NR4A3 on the expression of genes involved in inflammatory response and complement activation as well as metabolism, referring to the known function of NR4A3 in the transcriptional regulation of inflammatory genes and metabolism[12,14,25]. Higher expression of the glycolytic enzyme ENO3 in human AciCC tissues and our mouse salivary gland cell model with stable NR4A3 overexpression thus supports an additional positive effect of NR4A3 on cell metabolism in AciCC.

NR4As belong to the immediate-early genes, a group of genes (mostly transcription factors) whose expression is rapidly but transiently induced by external signals within minutes to initiate a rapid but transient cellular response to environmental stimuli[26–28]. Consistently, in a human parotid gland tissue sample we observed both activating and repressive chromatin marks in the promoter region and first intron of NR4A3 indicative of a bivalent/poised promoter. Notably, rapid induction of NR4A3 expression through histone acetylation of upstream regulatory elements has been reported in vascular smooth muscle cells[29]. The high frequency and specificity of the [t(4;9)(q13;q31)] aberration in AciCCs together with the consistent and specific upregulation of NR4A3, but not the neighboring genes, strongly suggests that NR4A3 is indeed the oncogenic driver in AciCCs. A recent study using transcriptomic sequencing analysis reported on a gene fusion on the mRNA level involving the SCPP gene cluster member *Histatin 3* (*HTN3*) and the gene *Myb/SANT DNA Binding Domain Containing 3* (*MSANTD3*) in one case of AciCC[9]. In the current study, we could not find any coding chimeric fusion transcripts at the mRNA level, even though one case (AciCC2) displayed an intragenic breakpoint involving the SCPP gene cluster member *Follicular Dendritic Cell Secreted Protein* (*FDCSP*). This, however, did not result in a chimeric fusion transcript. Instead, we suggest that the highly active chromatin marks we have observed at the SCPP salivary gland gene cluster enable upregulation of NR4A3 through inappropriate regulation of the *NR4A3* gene. Indeed, the comparison of 15 AciCCs with the [t(4;9)(q13;q31)] aberration revealed a relatively broad distribution of the 4q13 breakpoints among the SCPP salivary gland gene cluster and no minimal overlapping genomic region was present across all cases, excluding the possibility of a single shared genomic region influencing the expression of

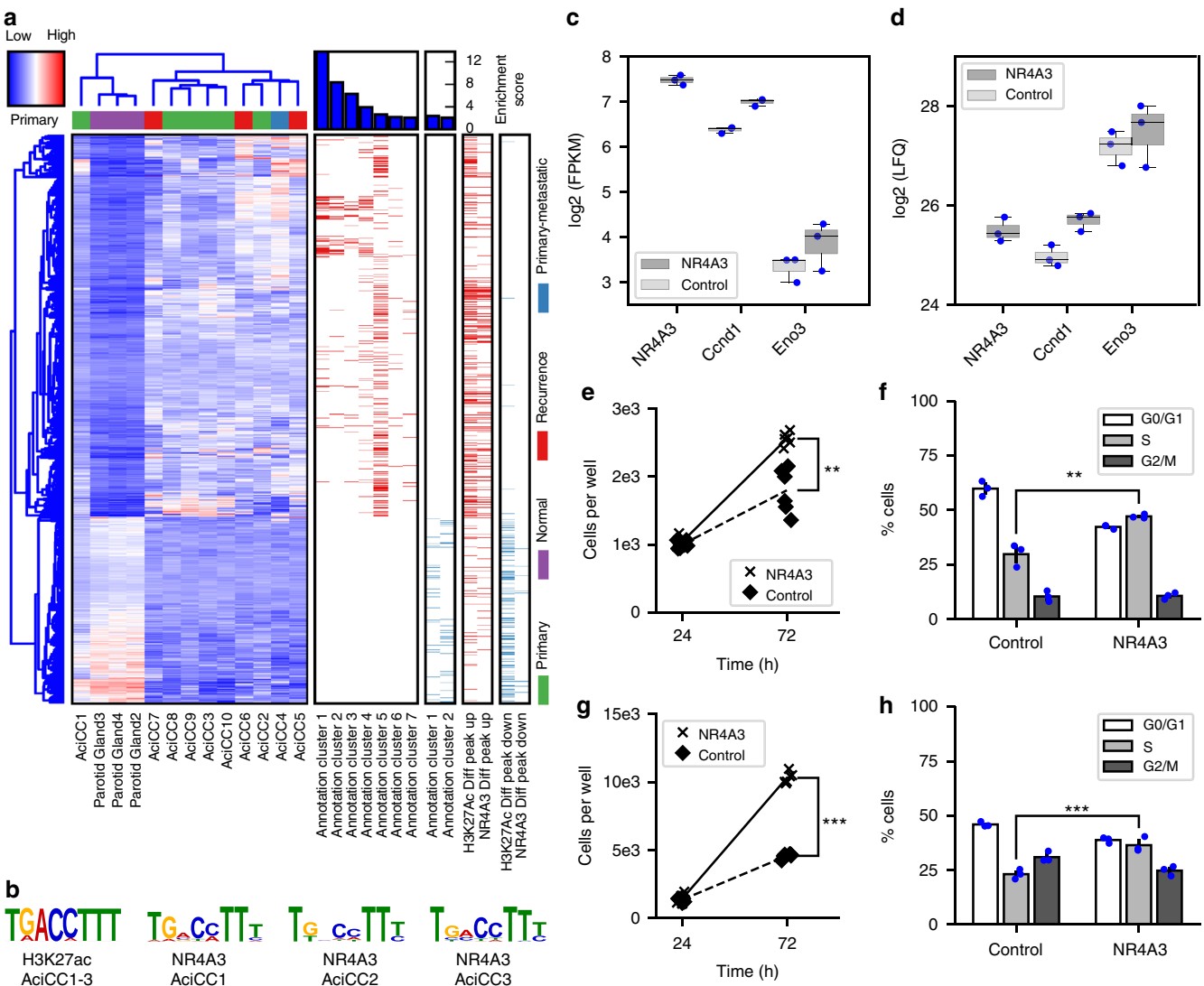

**Fig. 6** Functional impact of NR4A3 upregulation in AciCCs. **a** Hierarchical clustering of ten AciCCs and three normal parotid gland samples based on significantly up- and downregulated genes (deSeq2[47]; FDR = 0.01). Each row represents a gene, and each column represents a sample, with color-coding of gene expression levels. With the exception of one case, the tumors cluster separately from the normal tissues. Functional enrichment analysis reveals seven annotation clusters among upregulated genes, and two annotation clusters among downregulated genes. Genes associated with annotation clusters and differential H3K27ac and NR4A3 peaks are indicated by bars. Red and blue bars encode up- and downregulated genes associated with functional annotation clusters (left), and genes associated with up- and downregulated H3K27ac and NR4A3 peaks (right), respectively. **b** The NBRE motif is the only significantly enriched known motif at sites with upregulated H3K27ac peaks in the tumor samples (HOMER[54]; P < 0.01). De-novo motif analysis of NR4A3 transcription factor ChIP-seq peaks in the tumor samples reveals the same NBRE motif in all three samples. **c** mRNA expression and **d** protein levels of NR4A3, Ccnd1, and Eno3 based on RNA sequencing and mass spectrometry analysis, respectively, in immortalized mouse submandibular gland cells transduced with NR4A3 or red firefly luciferase (RedFF) negative control. Box-plots and individual values for three replicates are shown (Box-plot center line: median; bounds of box: 25 and 75 % quantiles; whiskers: minimum and maximum values). **e**, **f** immortalized mouse submandibular gland cells and **g**, **h** human mammary MCF10A cells with and without overexpression of NR4A3 open reading frames were seeded in microscopy plates and incubated for the indicated times. **e**, **g** Cell numbers were counted using high-content screening microscopy after staining with Hoechst-33258. Individual values of six replicates for each time point are presented. **f**, **h** For cell cycle analysis, cells were labeled with BrdU and 7-AAD and analyzed by FACS. Averages and standard deviations as well as individual values of three replicates are shown. Two-tailed Student's t-test; **P < 0.01, ***P < 0.001. Source data for Figs. 1a and 1c–h are provided as a Source Data file

NR4A3 in all samples. In contrast, we identified three distinct subgroups according to the location of the breakpoints within specific genomic regions, and different mechanisms of regulatory influence of 4q13 genomic regions are possible. We focused on the largest group of tumors and functionally validated a distal enhancer located within a 4q13 genomic region included in the translocation in the majority of the cases, thus indirectly presenting reasonable evidence that indeed the 4q13 enhancer activity is the major factor enabling NR4A3 upregulation. In

conclusion we suggest that our comprehensive patient and in vitro data provides strong evidence that the recurrent [t(4;9) (q13;q31)] aberration enables upregulation of NR4A3 through enhancer hijacking and constitutes the initial oncogenic driver event in AciCC. This concept mirrors structural genomic variations in other tumors that have been shown to disrupt spatial organization and activate oncogenes through inappropriate influence of regulatory regions or insulator dysfunction[6–8]. In view of pharmacologic compounds showing antagonistic effects

on NR4A family members[30], our observations could pave the way for the development of novel treatments through modulation of NR4A3 activity in AciCC in the future.

## Methods

**Patient cohort and informed consent.** Ten patients with diagnosis of AciCC of the salivary glands were enrolled in the initial study cohort, and their tumor material underwent extensive comprehensive genetic and epigenetic profiling. Eight patients gave written informed consent to participate in this study. In the two other patients who had deceived before the study start, archived frozen tumor material was analyzed and written informed consent was given by family members. The study was approved by the local ethics committee of the Friedrich-Alexander University Erlangen-Nuremberg (355_17 Bc). The remaining tumor samples used for FISH, hybrid capture next generation sequencing and immunohistochemistry were collected from the archives of the Institute of Pathology in accordance with ethical guidelines for the use of retrospective tissue samples by the local ethics committee of the Friedrich-Alexander University Erlangen-Nuremberg (ethics committee statements 24.01.2005 and 18.01.2012).

**Immunohistochemistry.** After formalin-fixation and paraffin-embedding of tumor material, a tissue microarray was conducted with two 2 mm cores per tumor using an automated tissue microarrayer (TMA Grandmaster, 3DHistech, Hungary). Immunohistochemistry was performed on 3 µm sections freshly cut from the TMA block using a fully automated staining system ("Benchmark XT System", Ventana Medical Systems Inc, 1910 Innovation Park Drive, Tucson, Arizona, USA) and monoclonal antibodies against human NR4A3 (1:100 dilution, sc-393902 (H-7), Santa Cruz Biotechnology, INC., Heidelberg, Germany) against human Cyclin D1 (1:50 dilution, clone SP4, Zytomed, Berlin, Germany), and rabbit polyclonal against human Enolase 3 (Beta, Muscle) (1:2000 dilution, HPA000793, Sigma-Aldrich Chemie GmbH, Steinheim, Germany). Cases that were negative for NR4A3 on the TMA slides were subjected to staining of freshly cut whole tissue slides and were re-evaluated. Semi-quantitative scoring for NR4A3 and Enolase 3 immunostaining was evaluated as 0, negative; 1+, weak/focal staining; 2+, intermediate/diffuse staining, and 3+, strong and diffuse staining. The percentage of positively stained nuclei for Cyclin D1 was counted. For NR4A3 and Cyclin D1, only nuclear staining was considered positive, while for Enolase 3, cytoplasmic staining was considered positive. Representative images of tumor and normal parotid gland areas were obtained using an Imager.A2 microscope and AxioCam MRc at ×400 magnification (Carl Zeiss Microscopy GmbH, Jena, Germany) with AxioVs40x64 V 4.9.1.0 software.

**Fluorescence in situ hybridization (FISH).** FISH was performed on freshly cut sections from tumor tissue blocks using the ZytoLight SPEC NR4A3 Dual Color Break Apart Probe (ZytoVision GmbH, Bremerhaven, Germany) with standard protocols according to the manufacturer's instructions. Fifty tumor cells were visually inspected using a fluorescence microscope and a cut-off of >20% tumor cells with aberrant signals was used to classify samples as NR4A3-rearranged. All cases were independently scored by two pathologists (F.H., A.A.) and revealed concordant results. The image presented in Supplemental Figure S2 was obtained with Leica SP5 II, Software Revision 2.6.3.1873 using a glycerol objective lens (×63, NA 1.3), detection with PMTs. Nyquist criterion matched for xy and z for all three channels, super critical pinhole with 0.8 Airy Units each channel. Stacks were post processed using Huygens Prof. Vers. 18.04 with deconvolution parameters: CMLE algorithm, SNR:20, 40 iterations.

**Whole genome sequencing.** DNA from AciCC tumors and from normal salivary glands was isolated from fresh-frozen tissues using DNeasy Tissue kits, and from blood of the same patients using the DNeasy Blood Kit (all Qiagen, Hilden, Germany). DNA-quality and quantity were assessed using a 2200 TapeStation (Agilent, Waldbronn, Germany). A total of 1 µg DNA was fragmented (E220; Covaris, Woburn, MA, USA) from each sample and size selected to 300 base pairs (bp) using an E-Gel system (Life Technologies, Darmstadt, Germany). DNA sequencing libraries were prepared using the TruSeq NanoDNALibrary PrepKit (Illumina, San Diego, CA, USA) following the manufacturer's instructions. Paired-end sequencing (2 × 150 bp) was performed using one lane of a HiSeqX (Illumina) for every sample.

Reads were aligned against the phase II reference sequence of the 1000 Genomes Project including decoy sequences d5 (ftp://ftp.1000genomes.ebi.ac.uk/vol1/ftp/technical/reference/phase2_reference_assembly_sequence/hs37d5.fa.gz) using bwa mem (version 0.7.8)[31] with default parameters, except for invoking -T 0. After alignment duplicates were marked by applying either Picard (version 1.125) or sambamba (version 0.5.9)[32]. The use of sambamba was necessary, since Picard did not finish for two samples. Genome coverage depth was calculated by custom Perl scripts, described elsewhere[33]. We called SNVs using our in-house, samtools-mpileup based variant calling pipeline[33,34], that was also used in the ICGC PanCancer[35] project. The workflow can be accessed via the Dockstore webpage: https://dockstore.org/containers/quay.io/pancancer/pcawg-dkfz-workflow.

In brief the workflow first determined variants in the tumor samples and compared these to a matched control to distinguish between somatic and non-somatic calls. The raw calls were further annotated using several publicly available data tracks, such as 1000 Genome variants, single nucleotide polymorphisms database (dbSNP), repeats, and other elements. For annotation of functional impact we used Annovar[36] and further assessed for the variants confidence. Raw calls for indels were obtained from Platypus (v.0.7.4)[37]. Annotation and confidence assessment was done similar to SNV processing. Calculation of genomic translocations was done using the CREST software[38]. In short, we first extracted soft clipped parts of the aligned reads, and then applied crest following the default parameters. In order to filter out germline events we used the -g option allowing for a comparison with a control bam file. In case we had no matched control a fake control was used to filter noise. The resulting translocation events were then annotated using annovar[36]. For visualizing the genomic context of the translocations we defined common TAD regions on the basis of 29 published HiC maps[10]. In summary, we first segmented the genome into 40 kb bins, and counted how many of the 29 samples contained a TAD boundary within each bin and its directly neighboring bins. If the count exceeded a number of 2 we checked if the next bin and its direct neighbors also exceeded a number of two and extended the boundary region by the end of the next bin. This was repeated until a subsequent bin did not fulfill the count criterion anymore, and then the TAD boundary position was defined as the boundary region start position plus the length of the boundary region divided by two. Furthermore, we used the HiC map of the mesoendoderm cell line replicate two for visualizing fine grained TAD structures. Allele-specific copy-number alterations were detected using ACEseq (allele-specific copy-number estimation from WGS)[39]. ACEseq determines absolute allele-specific copy numbers as well as tumor ploidy and tumor cell content based on coverage ratios of tumor and control as well as the B-allele frequency (BAF) of heterozygous SNPs. SVs called by CREST[38] are incorporated to improve genome segmentation.

**Whole genome bisulfite sequencing.** DNA from AciCC tumors were isolated from fresh-frozen tissue using DNeasy Tissue kits and from blood of the same patients using the DNeasy Blood Kit (all Qiagen, Hilden, Germany). DNA-quality and quantity were assessed using a 2200 TapeStation (Agilent, Wald-bronn, Germany). A total of 100 ng DNA was fragmented (E220; Covaris, Woburn, MA, USA) from each sample and size selected to 300 bp using an E-Gel system (Life Technologies, Darmstadt, Germany). DNA sequencing libraries were prepared using the Accel-NGS® Methyl-Seq DNA Library Kit (Swift Biosciences, Ann Arbor, MI, USA) following the manufacturer's instructions. Paired-end sequencing (2 × 150 bp) was performed using one lane of a HiSeqX (Illumina) for every sample. Raw paired-end sequence reads were trimmed using Trimmommatic (v. 0.36)[40] with default parameter values to eliminate remnants of the sequencing adapters. Read pairs for which both mates survived the trimming were aligned to the GRCh37 assembly of the human genome reference sequence using bwameth (v. 0.2.0)[41], a wrapper of the bwa mem[1] alignment algorithm suited for bisulfite sequencing data. Redundant reads emerging during PCR amplification were removed using MarkDuplicates tool of the Picard suite (v. 2.5.0-1) (http://broadinstitute.github.io/picard). Methylation states of the CpG cytosines were called with MethylDackel (v. 0.2.1) (https://github.com/dpryan79/MethylDackel.git). During methylation calling 10 base pairs were disregarded from 5'-ends of both read mates to eliminate methylation bias artefacts.

**RNA sequencing.** RNA was isolated from AciCC tumors and from cell lines derived from mouse SG (see below) and stably transfected with NR4A3 or red firefly luciferase ORFs, respectively, using the RNeasy Plus Mini Kit (Qiagen). Quality of RNA was analyzed using Agilent Bioanalyser 2100 (Agilent). A volume of 1 µg total RNA was fragmented to a median length of 300 bp (Covaris). Library preparation for RNA-seq was performed using the TruSeq RNA v2 kit for tumor RNAs and the Illumina TruSeq stranded total RNA protocol (Illumina) and NEBNext Multiplex Oligos (New England Biolabs) for cell line RNAs, following the manufacturers' instructions. Sequencing was with paired-end sequencing with 2 × 100 bp HiSeq 4000 and 2 × 125 bp HiSeq2000 v4 technologies (Illumina) for tumor and cell line samples, respectively. Three to four libraries were pooled per lane. Reads were aligned against a STAR reference index generated from the 1000 genomes phase 2 assembly, gencode 19[42] gene models and for a sjdbOverhang of 200. This action was followed by a 2 pass alignment using the STAR aligner (version 2.5.2b)[43]. The alignment call parameters were:

*--sjdbOverhang 200 --runThreadN 8 --outSAMtype BAM Unsorted SortedByCoordinate --limitBAMsortRAM 100000000000 --outBAMsortingThreadN = 1 --outSAMstrandField intronMotif --outSAMunmapped Within KeepPairs --outFilterMultimapNmax 1 --outFilterMismatchNmax 5 --outFilterMismatchNoverLmax 0.3 --twopassMode Basic --twopass1readsN -1 --genomeLoad NoSharedMemory --chimSegmentMin 15 --chimScoreMin 1 --chimScoreJunctionNonGTAG 0 --chimJunctionOverhangMin 15 --chimSegmentReadGapMax 3 --alignSJstitchMismatchNmax 5 -1 5 5 --alignIntronMax 1100000 --alignMatesGapMax 1100000 --alignSJDBoverhangMin 3 --alignIntronMin 20 --clip3pAdapterSeq AGATCGGAAGAGCACACGTCTGAAC TCCAGTCA --readFilesCommand*

Other parameters were as default. Duplicate marking and index generation on the resulting alignment, and chimeric bam files was performed using sambamba (version 0.6.5)[32]. For sorting we used samtools sort (version 1.3.1)[44]. Quality control analysis was done by applying the samtools flagstat[14] command, and the rnaseqc tool (version 1.1.8)[45] with the 1000 genomes phase 2 assembly and gencode 19 gene models as input. Depth of Coverage analysis for rnaseqc was turned off. Featurecounts (version 1.5.1)[46] was used to perform gene specific read counting over exon features based on the gencode 19 gene models. Both reads of a paired fragment were used in a strand-unspecific manner for counting and the quality threshold was set to 255 (which indicates that STAR found a unique alignment). A custom script was used to calculate FPKM and TPM expression values. For total library abundance calculations, all genes on chromosomes X, Y, MT, as well as rRNA and tRNA genes were omitted as they are likely to introduce library size estimation biases. We analyzed our RNA-sequencing samples for differential expression (tumor vs. normal) by applying the DeSeq2 R package[47] on per sample read counts as determined by the featurecounts software[17]. We removed all genes that had no read counts among our samples from further analysis. Furthermore, we determined the regularized log transformed read count matrix for visualization of the data. Differential analysis was done on 10 tumor samples vs. 3 normal samples. We only considered differentially expressed genes with a FDR smaller or equal to 0.01. Pathway analysis was performed using the DAVID web service[48] on the HUGO gene symbols of differentially expressed genes (FDR <= 0.01). We included the following gene sets into our analysis: From functional categories we used COG_ONTOLOGY, UP_KEYWORDS, and UP_SEQ_FEATURE. From gene ontology we used GOTERM_BP_DIRECT, GOTERM_CC_DIRECT, and GOTERM_MF_DIRECT, and from pathways we used KEGG_PATHWAY. For annotation clustering we set the confidence threshold to an EASE score of 0.1, and for the annotation chart we set the EASE score threshold to 0.01. We repeated the analysis independently for up- and downregulated genes with respect to the tumor. For the detection of gene fusions we applied deFuse[49] and the tool arriba (https://github.com/suhrig/arriba) using default parameters.

**ChIP sequencing**. Illumina sequencing libraries were prepared from the ChIP and Input DNAs by the standard consecutive enzymatic steps of end-polishing, dA-addition, and adaptor ligation using the Apollo 342 from Wafergen Biosystems. After a final PCR amplification step, the resulting DNA libraries were quantified and sequenced on Illumina's NextSeq 500 (75 nt reads, single end). ChIP-sequencing reads were aligned against the 1000 genomes phase 2 assembly using bwa aln (version 0.7.12)[1] with default parameters. Afterwards duplicate removal was performed by samtools rmdup (version 0.1.19)[44] with parameter -s. Since the 5′-ends of aligned reads (="tags") represent the end of ChIP/IP-fragments, the tags were extended in-silico (using custom software) at the 3′-ends to a length of 150–250 bp, depending on the average fragment length in the size selected library (normally 200 bp). To identify the density of fragments (extended tags) along the genome, the genome was divided into 32 bp bins and the number of fragments in each bin was determined. This information was stored in bedGraph files and used for visualization. For finding regions of local enrichment in ChIP-seq read density, so-called peaks, we used MACS (version 2.1.0)[50] with parameters effective genome size = 2.7e + 09, band width = 200, model fold = [5, 50], p-value cutoff = 1e-07, and broad calling = off, or SICER (version 1.1)[51] with parameters window size = 200, fragment size = 200, effective genome size as a fraction of hs37d5 = 0.86, gap size = 0, evalue for identification of candidate islands that exhibit clustering = 1000, FDR = 1e-10, depending on the expected breadth of the peaks. Before applying peak calling, we normalized all samples for a particular ChIP/IP by sampling down all samples to have the same tag counts as the sample with the lowest tag count. Afterwards, we removed peaks residing in ENCODE black-listed regions. Since MACS was particularly made for finding narrow peaks we used it for peak calling on ChIP-Seq data of H3K27ac, H3K4me3, CTCF, and NR4A3. SICER was only used for broad H3K27me3 signals. For the identification of super enhancers we used a proprietary software implemented by Active Motif that gives very similar results as the ROSE super-enhancer software[52,53]. In a first step, MACS peaks of H3K27ac were merged if their distance was equal or less than 12,500 bp. In a second step, the merged peak regions with the strongest signals (top 5%) were identified as super enhancers. For defining differential peaks between tumor and normal samples we first calculated for each histone mark extended regions for which at least one of the samples contains a peak (i.e., the union of peaks). Afterwards, we calculated for each sample and each extended peak interval the average fragment density among all bins within the region. This enabled us to determine the log2 fold change of average fragment density between tumor samples and the normal sample. We then defined the intervals with log2 ratio smaller or equal than −2 as downregulated in tumors and the intervals with log2 ratio greater or equal than 2 as upregulated in tumors. Detection of enriched TFBS motifs in differential peaks was performed using the HOMER software[54]. The perl script findMotifsGenome.pl was applied using default parameters except for -size given and -keepFiles. As background we used the complete set of extended regions per histone mark. We scanned for CTCF motifs within the individual samples' peaks using the motif package of the BioPython module[55]. The position weight matrix (PWM) of CTCF was obtained from the HOCOMOCO database[56]. To avoid overfitting of the PWM we introduced pseudocounts

according to an approximate CG content in the human genome of about 40%. As scanning background we used a global distribution of A,T = 30% and C,G = 20%. Scanning of the motifs according the PWM was done using a precision of 10e4 and a FPR of 10e-4. De-Novo motif search on NR4A3 peak regions was performed using the MEME[57] web interface (http://meme-suite. org/). First we extracted for each tumor sample the sequences of the 1000 top ranking peak regions, based on the maximum signal value within the peak. On these the MEME algorithm was applied using the default parameters, except for Minimum Motif With = 8, Maximum Motif Width = 20, Minimum Sites per Motif = 100, and Maximum Sites per Motif = 300.

**Hybrid capture sequencing**. DNA was extracted from freshly cut slides from formalin fixed and paraffin embedded tumor blocks after manual microdissection of tumor areas using the Qiagen QIAamp DNA FFPE Tissue Kit. Libraries were prepared with a customized hybrid capture based approach using probes directed against a ~ 315 kbp region upstream of the NR4A3 TSS and the SureSelectXT HS Target Enrichment System (Agilent). After quality control, libraries were sequenced on NextSeq 550 (Illumina). Alignment and variant calling was done with the SureCall software for the customized hybrid capture approach (Agilent). Translocations were filtered by a quality score greater or equal to 255, and by a minimal support of the translocation of 15 reads. The remaining translocations were visually inspected using the Integrated Genomics Viewer (IGV) software[58,59] and further filtered.

**Dual luciferase reporter assays**. Genomic regions-of-interest were PCR-amplified from commercial human DNA (Roche, catalog no. 11691112001) using the Q5 kit (New England Bioloabs), cloned into reporter vector pGL4.23 (Promega, catalog no. E841A) and sequence verified by Sanger sequencing (GATC Biotech). About 8000 293FT cells were transfected using the TransIT LT1 transfection reagent (Mirus, catalog no. MIR 2304) with 50 ng plasmid mix consisting of 20 ng of the reporter construct or empty vector pGL4.23, 10 ng of the pRL-TK Renilla luciferase reporter vector (Promega, catalog no. E2241) and, as stuffer DNA, 20 ng of a pGL3 (Promega) deletion derivative with a non-functional luciferase gene. The dual luciferase readout was performed with nearly confluent cells using the SpectraMax M5 (Berthold, Bad Wildbad, Germany); the firefly luciferase reporter signals were normalized to the renilla luciferase transfection control signals. Mean values were calculated from six parallel measurements per transfection (technical replicates), and at least three independent transfections were performed per construct (biological replicates). Statistical evaluation of differences between mean values and empty vector (values of normalized signals set to 1) was performed by one-sample *t*-test.)

**Overexpression of NR4A3 in mouse SG cells**. Three-week-old female C57BL/6N mice were obtained from Envigo CRS GmbH (Rossdorf, Germany) maintained under standard housing conditions in the DKFZ animal facility for one week. After acclimatization the mice were euthanized by $CO_2$ inhalation for isolation of SGs. Performance was in accordance with European guideline 2010/63/EU and the guidelines of the local authorities of the DKFZ. SG cells were prepared as described elsewhere[60] with minor modifications. In brief: SGs were excised and transferred to ice-cold phosphate buffered saline (PBS; Thermo Fischer Scientific/ Gibo-BRL, Germany) enriched with 5% fetal calf serum (FCS; Thermo Fischer Scientific/ Gibo-BRL, Germany) and 1% penicillin-streptomycin (Thermo Fischer Scientific/ Gibo-BRL, Germany). Residual blood was removed by two washing steps in ice-cold PBS. SGs were minced in Dulbecco's modified Eagle's medium (DMEM, Thermo Fischer Scientific/ Gibo-BRL, Germany) supplemented with 10% FCS. Minced SG tissue was enzymatically digested for 1 h at 37 °C in 5 mL DMEM supplemented with collagenase type-1 (15 mg; Merck, Germany), hyaluronidase type I-S (5 mg; Merck, Germany), 10% FBS, and DNase I (1 U/mL; Qiagen Germany), and vortexed in 10 min intervals. Cells and tissue fragments were separated by filtering through a 70 cm cell strainer (BD Biosciences). Cells were washed twice in ice-cold PBS (500 g/5 min) and the supernatant of each wash was discarded. The SG cell pellet was suspended in epidermal keratinocyte medium (CnT-07; CELLnTEC Advanced Cell Systems, Bern, Switzerland) containing 0.01 mg/mL human recombinant epidermal growth factor (EGF; Merck, Germany), 2.5 mg/mL penicillin-streptomycin, and 0.1 mg/mL CT (Merck, Germany) in a concentration of $10^6$ cells/mL and $10^7$ cells in total were seed/10 cm² dish coated with 0.1% gelatin (Merck, Germany) and incubated at 37 °C and 5% $CO_2$. Medium was changed every two days and the cells were cultured in epidermal keratinocyte medium containing the respective antibiotics. At 80% density the SG cells were passaged 1:4 using Accumax (Thermo Fischer Scientific, Germany) for detachment and reseeded on gelatin coated plates.

To immortalize primary mouse SG cells, the HPV-E6-E7[61] ORF flanked by attL recombination sites was cloned into a pMK-RQ plasmid also carrying a kanamycin selection marker (Thermo Fischer Scientific). The construct was shuttled by Gateway recombination technology (Thermo Fischer Scientific) in a lentiviral expression vector fusing a c-terminal internal ribosomal binding site (IRES), and harboring a neomycin resistance marker. To analyze the effects of NR4A3 overexpression, the (ORF) of human NR4A3 (NM_173200) flanked by attL recombination sites was synthesized (GeneArt, Germany) and cloned into the pMK-RQ vector (Thermo Fischer Scientific). For normalization, the ORF of red

firefly luciferase (Thermo Fischer Scientific) was amplified using gene specific primes with flanking attB recombination sites. The resulting PCR-product was recombined in pDONR221 (Thermo Fischer Scientific) and sequence validated. The ORFs of NR4A3 and red firefly luciferase were shuttled into a lentiviral expression vector adding a c-terminal IRES sequence coupled to a puromycin resistance gene by Gateway recombination.

For generation of lentiviral particles, 293FT cells (Thermo Fischer Scientific, Germany) were co-transfected with the lentiviral expression constructs and 2nd generation viral packaging plasmids VSV.G (Addgene #14888) and psPAX2 (Addgene #12260). 48 h after transfection, virus containing supernatant was removed and cleared by centrifugation (5 min/500 g). The supernatant was passed through a 0.45-μm filter to remove remaining cellular debris. SG cells were immortalized by transduction with HPV-E6-E7[61] lentiviral particles at 70% confluency in the presence of 10 μg/mL polybrene (Merck, Germany). 24 h after transduction remaining viral particles were removed and immortalized cells were selected in the presence of 500 μg/mL neomycin. For functional analysis immortalized SG cells were transduced with lentiviral particles for overexpression of either NR4A3 or red firefly luciferase control in the presence of polybrene as described above. Stably transduced cells were obtained by selection with 500 μg/mL neomycin and 1 μg/mL puromycin (Merck, Germany). The cells were tested for mycoplasma contamination on a regular basis.

**Stable transduction of human mammary gland epithelial MCF10A cell lines with NR4A3**. MCF10A cell line was cultured in complete growth medium under standard culture conditions (37 °C; 5% $CO_2$). Cells were authenticated upon arrival (ATCC CRL-10317), as well as after completion of the study (Multiplexion, Heidelberg), and tested for mycoplasma infection on a regular basis. Cells were transduced with lentiviral NR4A3 overexpression particles as described above at 70% confluency. Twenty four hours after transduction remaining viral particles were removed and the cells were selected with 1 μg/mL puromycin (Merck, Germany). Single clones were picked and tested for expression of NR4A3 by qRT-PCR. Clones showing high and no detectable NR4A3 overexpression, respectively, were selected for functional experiments testing effects of NR4A3 (Supplemental Figure 10, Source Data file).

**Functional analysis of NR4A3 activities on proliferation and cell cycle**. Proliferation of SG or MCF10A cells with or without overexpression of NR4A3 was assessed by cell counting at different time points after seeding. A total of 1500 cells were seeded into each well of 96-well microscopy plates (Greiner, Germany) and grown for 24 h and 72 h at 37 °C with 5% $CO_2$. Hoechst-33258 was added to the cells at a final concentration of 1 μM and incubated for 30 min before measurement sing an ImageXpress Micro high-content screening microscope (Molecular Devices). Images were analyzed with the built-in analysis app "Count Nuclei". Average and standard deviation of six replicates for each time point were calculated. The two-tailed Student's t-test was used for statistical analyses of the difference in cell numbers after 72 h.

To analyze cell cycle profiles of SG or MCF10A cells with and without overexpression of NR4A3, $1.5 \times 10^5$ cells/well were seeded into 6-well plates. Twenty four hours after seeding, the cells were starved for 24 h followed by 24 h treatment with full growth medium. Then, the cells were incubated with BrdU for 2 h before fixation with Cytofix/Cytoperm (BD Biosciences, Heidelberg, Germany) according to the manufacturer's recommendations. BrdU was stained with a FITC-conjugated anti-BrdU antibody (BD Biosciences, Heidelberg, Germany) and DNA was stained with 7-AAD. Samples were analyzed using a FACSCalibur (Becton-Dickenson) and their distributions in the distinct cell cycle phases G0/G1, S and G2/M phases was assessed based on their fluorescence intensity for both dyes as detailed in Supplementary Figure 11. The two-tailed Student's t-test was used for statistical analyses of the differences in the fraction of cells in S-phase.

**Real-time PCR**. Cells were seeded into transparent 6-well plates and RNA was isolated with mRNeasy Mini kit (Qiagen, Hilden, Germany) the following day at 80% confluency. mRNA quantification was performed with TaqMan®-based qRT-PCR and UPL probes (Roche Diagnostics GmbH, Mannheim, Germany). Primer sequences were left-ggaggggggttgaggtgtt and right-gtgtgcacttttattggtctcaa for Actb using UPL#71, left-tttctttccagagtcatcaagtgt and right-tgactccagaagggcttcaa for Ccnd1 using UPL#72, left-gaggggggtgaactgacactg and right-gagtcttctccacccgaaga for Eno3 using UPL#67, left-gcccaatacgaccaaatcc and right-agccacatcgctcagacac for GAPDH using UPL#60, and left-ctcaacacccagagatcttgatta and right-gtagaattgttgcacatgctcag for NR4A3 using UPL#66. Data were processed with SDS 2.2.2 software (Thermo Fisher Scientific, Waltham, USA) and quantifications were performed with 2-ΔΔCT method. The two-tailed Student's t-test was used for statistical analyses.

**Immunoblotting**. Protein abundance was quantified with western blotting. Cells were seeded into 6-well plates and proteins were isolated the following day at 80% confluency. Primary antibodies used were Anti-Actin (clone C4) (MP Biomedicals GmbH, Eschwege, Germany) for β-Actin (1:10,000 dilution) and SC-718 (Santa Cruz Biotechnology, Inc., Heidelberg, Germany) for cyclin D1 (1:200 dilution). Bands were visualized with secondary IRDye®680 and IRDye®800-conjugated

antibodies and membranes were scanned with Odyssey scanner and analyzed with Odyssey 2.1 (LI-COR, Lincoln, NE, USA).

**Reporting summary**. Further information on experimental design is available in the Nature Research Reporting Summary linked to this article.

## Data availability

All sequencing data supporting the findings of this study have been uploaded to the European Genome-Phenome archive (http://www.ebi.ac.uk/ega/), and are available via the accession number EGAS00001002795 All processed data that support the findings of this study are available in ZENODO with the identifier [10.5281/zenodo.1483691][62]. A reporting summary for this Article is available as a Supplementary Information file. The source data underlying Figs. 1b, c, e, 5, 6a, c–h, and Supplementary Figures 4, 6, 7, 8, 10 are provided as Source Data file. All other data are available from the corresponding authors upon reasonable request.

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

## Acknowledgements

We thank Bishaka Roy and Andrea Waxmann from the sequencing and Bernd Hessling and Martin Schneider from the protein analysis units of the DKFZ Genomics and Proteomics Core Facility for performing excellent services. We thank Christa Winkelmann and Susanne Blank from the Institute of Pathology for excellent contribution to immunohistochemical and in situ analyses and Heike Wilhelm for excellent technical assistance with molecular and cellular biology experiments. We thank the DKFZ-Heidelberg Center for Personalized Oncology (DKFZ-HIPO) and Dr. Carl Hermann, Department of Theoretical Bioinformatics, for helpful expertize. Fluorescence microscopy/image analysis was performed with the support of the Optical Imaging Centre Erlangen (OICE) and the excellent expertize of Dr. Ralf Palmisano. Bioinformatic analysis of hybrid capture based sequencing of NR4A3 upstream region was performed with the helpful support of Dr. Michael Walter, Agilent Technologies Sales and Services GmbH.

## Author contributions

M.Bi., N.I., P.L., K.K. and R.E. performed or supervised sequencing data analyses. R.W. and B.K. established cell line models. C.K., A.B., A.W. and S.W. performed in vitro assays and performed data analysis. D.W. and M.Ba. performed and analyzed dual luciferase reporter assay. S.K.M., C.S., M.H., R.F., H.I. and A.H. collected specimens. F.H., E.A.M. and A.A. performed immunohistochemical and in situ analyses. F.H., M.Bi. and E.A.M. performed and analyzed hybrid capture next generation sequencing analyses. F.H., M.Bi., R.W., C.P., S.W. and A.A. interpreted the results. F.H., M.Bi., R.W. and A.B. generated figures and tables. F.H., M.Bi., C.P., S.W. and A.A. conceived the study and wrote the manuscript. All authors participated in discussions and interpretation of the data and results.

## Additional information

**Competing interests:** The authors declare no competing interests.

