## [Peer Review File · Nature Communications]

Reviewer #1 (Remarks to the Author):

The manuscript by Haller et al. describes the novel finding that recurrent translocation of SCPP locus and NR4A3 might lead to upregulation of NR4A3, which might drive the oncogenic pathways in AciCC. The authors show differentially over expressed genes with active histone marks regions are enriched for NR4A target motif sites, adding more support to the hypothesis that oncogenic expression of many genes is driven by NR4A3 activity. Overall the genetic data is very convincing, the findings are novel, the manuscript is nicely written, which warrant publication.

The authors show that the translocation breakpoints in the SCPP locus are in active (H3K27Ac) regions, which juxtaposes this with NR4A3, and lead to upregulation of NR4A3. It is suggested that this is because of "super-enhancer" (SE) juxtaposition. However, in not every case there seems to be detectable SE activity (Fig. 2 and 3 purple bars). How are these SE on hockey-stick plots/ROSe analyses?

Further exploration to test if these regions are indeed SE may help. The genetic data and some of the ChIP-seq suggest an activation mechanism, even if there is no "SE" activity. There is a reduction in H3K27me3 (bivalency states). May be juxtaposed regular enhancer, not necessarily SE, is sufficient for activation of NR4A3 and drive the phenotypes. Are the tissue samples amenable to conformational capture assay, to directly show that the enhancer activity from the SCPP locus juxtaposes to the NR4A3 promoter. Experimental evidence for such interaction will be a key strengthening point. In the absence of data to directly show the enhancer activity by interaction, these caveats should be discussed.

Is it possible to grow AciCC in nude mice to test if enhancer-inhibitor strategy (such as BETi) will reduce tumor growth. This will further support the enhancer/SE hijacking hypothesis. Or, will knocking out NR4A3 block tumor growth in the nude mouse models?

Additional points:

- In text relating to Fig. 4, expression of human NR4A3 is suggested to increase endogenous mouse Nr4a3. However the data pertaining to Hprt and Nr4a3 expression is not available. Only human NR4A3 expression is shown, which is exogenously over-expressed.

- NR4A3 gene body methylation being significantly lower in two of the AciCC is correlated to sustained activated expression of NR4A3. This is not clear as most previous studies have suggested that gene-body methylation positively correlated with gene activity (increased expression associated with increased gene body methylation).

Reviewer #2 (Remarks to the Author):

This manuscript describes a genomic, transcriptomic, and epigenomic profiling of salivary acinic cell carcinomas (AciCC). WGS of 6 AciCCs revealed t(4;9)(q13;q31) translocations in all cases. There were no recurrent copy number alterations or mutations. The 4q13 breakpoints clustered within the secretory Ca-binding phosphoprotein (SCPP) gene cluster and the 9q31 breakpoints were located upstream of the orphan nuclear receptor gene NR4A3. RNA seq did not reveal any functionally fusion transcripts but showed significant upregulation of NR4A3 compared to normal parotid gland (NPG). No other genes up- or downstream of NR4A3 were overexpressed compared to NPG. The 4q13 breakpoints mapped to regions with enrichment of active chromatin marks associated with abundantly expressed salivary gland genes. Moreover, active chromatin marks from the SCPP cluster were located directly at the 9q31 breakpoint region upstream of NR4A3 in AciCCs. DNA methylation throughout the NR4A3 gene was significantly lower in two out of three AciCCs compared to normal salivary gland, consistent with activation of the gene. Mouse salivary gland cells stably transduced with a wildtype NR4A3 cDNA showed increased viability and cell

index compared to control cells, suggesting that NR4A3 has transforming potential in mouse cells. The authors conclude that the t(4;9) link active enhancer elements from highly expressed salivary genes in the SCPP cluster to the NR4A3 upstream region, resulting in activation of NR4A3 expression. It is also suggest that NR4A3 is an oncogenic driver in AciCC.

This is an ambitious study employing state-of-the-art methodology. The epigenetic analyses are comprehensive and solid. The study is, however, mainly descriptive but contains some new information. NR4A3 is a known oncogene previously shown to be activated by chromosome translocations in sarcomas. A major concern is the lack of convincing functional evidence proving that NR4A3 is an oncogenic driver in AciCC. The authors only present physical evidence of a t(4;9) by WGS in 6 cases and show that the translocation leads to overexpression of NR4A3. No breakpoints were identified in any genes in 4q13 or 9q31 and no functional fusion transcript were identified.

To substantiate the conclusion that NR4A3 is an oncogenic driver in AciCC the authors should knock-down the expression of the gene in human AciCC cells and study the effects of knock-down on proliferation, apoptosis etc. They should also study the transforming activities of NR4A3 in human salivary or mammary cells by analyzing the effects on proliferation, viability, spherogenesis, growth in soft agar etc.

Another weakness of the paper is that only one target gene of NR4A3 is studied (CCND1). Additional target genes should be analyzed in order to further characterize the molecular consequences of activation of NR4A3 in AciCC.

The SCPP gene locus (se Introduction lines 50-51 and elsewhere) is not a specific locus (gene) but a cluster of several different genes. It should therefore not be designated the "SCPP gene locus". In addition, information about the genes included in this cluster, their functions and role in salivary glands should be added.

The FISH image shown in Fig. 1D is difficult to interpret since the boundaries of individual nuclei are impossible to discern and therefore it is not possible to count the signals in these nuclei. The image should also include several nuclei showing the same FISH pattern.

The quality of the H&E sections of normal parotid gland and AciCC (Fig. 4A) could be substantially improved.

The authors should include more detailed information about the histology of their AciCC cases, that is whether they have a conventional histology or are cases with high-grade transformation. AciCC5 in Fig 1B looks heavily rearranged and could represent a case with high-grade transformation?

Fig. 2, showing the translocation breakpoints (TXs), is somewhat difficult to understand and needs to be explained or better illustrated. Do AciCC3-6 have inversions at the chromosome 4 bp (cf. Fig 3D)? Please provide a more detailed explanation of secondary events (e.g. inversion of the SCPP gene cluster) occurring at the breakpoint sites on 4q and 9q.

Fig 2, the last panel showing the gene body DNA methylation pattern. AciCC1 looks very similar to normal parotid gland and different from AciCC 2 and 3 (cf. Fig 4B where this case clusters with normal parotid gland). Does this case contain a large amount of normal parotid gland tissue and only limited amounts of tumor tissue?

It would be interesting to see the expression level of NR4A3 for each AciCC and to compare it to the type of translocation (where the breakpoints are located) and to the amount of activating chromatin marks in the vicinity of NR4A3.

I suggest that the authors use at least two software programs for detection of gene fusions.

I also suggest that the authors show cell counts over time instead of cell index and viability (Fig 4 I and G). There is a big difference between the effects presented in Fig 4G and 4I. The authors should also include a western blot of the transfected mouse cells and not only show the mRNA expression (Fig. 4H).

Reviewer #3 (Remarks to the Author):

The authors describe recurring rearrangements in acinic cell carcinoma (AciCC), and perform genome, transcriptome, and epigenomic profiling, implicating highly recurrent inter-chromosomal translocations in enhancer hijacking activating the oncogenic transcription factor NR4A3. Further gene expression and in-vitro analyses are presented corroborating a likely oncogenic role of NR4A3 in AciCC via Cyclin D1, including 2.5-fold upregulation of mouse *Ccnd1* mRNA, increased Cyclin D1 protein levels and increased cell index. This is an interesting contribution describing interesting and novel finding in a rare tumour entity. It is indeed likely that the active enhancers juxtaposed to NR4A3 can drive NR4A3 gene expression. This is a timely result that should be published soon.

My only major criticism pertains to the wording used by the authors. The authors state that 'the rearrangements translocate active enhancer regions from highly expressed salivary gland genes to the NR4A3 upstream region, resulting in upregulated 2 expression and nuclear accumulation of NR4A3.' The word "resulting" is an overstatement, and should be toned down, since the authors did not present direct experimental evidence for interaction between the enhancer elements and the NR4A3 promoter (as e.g. pursued in PMID:27869826 using 4-C sequencing). At least the remaining limitation that this test for direct interaction has not been pursued should be made clear in the Discussion.

Additional points:

- Figure 1: I agree NR4A3 seems to be a likely target here, but this figure needs to clarify a few aspects. Please indicate the p-value that '****' refers to. Please compute p-values for other genes, such as *SEC61B*, which might be significant too? What is the difference in fold change between NR4A3 and other genes. Please clarify whether there is sufficient evidence to rule out that the expression of other genes may have an additional role in AciCC.

- An analysis of common (cell type invariant) TADs (topologically associating domain structures) at the affected loci would be potentially useful for the readers. I assume the breakpoints will likely affect TAD boundaries with potentially insulating effect. It should be reassured though that there are no TAD boundaries between the active enhancer elements shown in Figure 3 and the target gene NR4A3. Datasets calling TAD boundaries present across distinct cell types could be used for such analysis.

Reviewers' comments:

Reviewer #1 (Remarks to the Author):

The manuscript by Haller et al. describes the novel finding that recurrent translocation of SCPP locus and NR4A3 might lead to upregulation of NR4A3, which might drive the oncogenic pathways in AciCC. The authors show differentially over expressed genes with active histone marks regions are enriched for NR4A target motif sites, adding more support to the hypothesis that oncogenic expression of many genes is driven by NR4A3 activity. Overall the genetic data is very convincing, the findings are novel, the manuscript is nicely written, which warrant publication.

> We thank the reviewer for his/her very positive comment, and appreciate his/her careful evaluation of our work.

The authors show that the translocation breakpoints in the SCPP locus are in active (H3K27Ac) regions, which juxtaposes this with NR4A3, and lead to upregulation of NR4A3. It is suggested that this is because of "super-enhancer" (SE) juxtaposition. However, in not every case there seems to be detectable SE activity (Fig. 2 and 3 purple bars). How are these SE on hockey-stick plots/ROSe analyses? Further exploration to test if these regions are indeed SE may help.

> We provide hockey-stick plots for all three AciCC samples as a novel Supplemental Figure S5, demonstrating that enhancer regions from the SCPP gene cluster in close proximity to the breakpoints rank among the strongest enhancer regions in two samples, and fulfill the criteria for super-enhancers in the third sample. This is now specifically described in the revised manuscript (page 8).

The genetic data and some of the ChIP-seq suggest an activation mechanism, even if there is no “SE” activity. There is a reduction in H3K27me3 (bivalency states). May be juxtaposed regular enhancer, not necessarily SE, is sufficient for activation of NR4A3 and drive the phenotypes.

> We agree with the reviewer that juxtaposition of a regular enhancer might be sufficient for activation of NR4A3, e.g. similar mechanisms were shown to drive TAL1, TLX1/TLX3, HOXA10 activation in T-ALL (reviewed in Belver & Ferrando, Nat Rev Cancer, 2016). We have now included the observation of reduced H3K27me3 marks in the revised manuscript (page 9).

Are the tissue samples amenable to conformational capture assay, to directly show that the enhancer activity from the SIPP locus juxtaposes to the NR4A3 promoter. Experimental evidence for such interaction will be a key strengthening point. In the absence of data to directly show the enhancer activity by interaction, these caveats should be discussed.

> As also suggested by reviewer 3, we aimed at providing circularized chromatin conformation capture combined with next generation sequencing (4C-seq) from patient tissues AciCC 1-3 using the NR4A3 breakpoint region as view point to clarify the possible interactions. Unfortunately, and as the reviewer was already concerned about, the amount of frozen tumor material that was left after the extensive ChIP-seq analysis that had been performed in preparation of the original submission was too small to yield sufficient data in the new 4C-seq analysis. The 4C-protocol we used is included at the end of the cover letter, the amount of chromatin we had available for this experiment was in the range of <10% of the recommended minimum. To still further confirm the enhancer activity within the SIPP gene cluster, we now provide novel data from an experiment where we used a dual luciferase reporter assay that indeed demonstrated enhancer activity within the SIPP gene cluster (page 9, Figure 5, Supplemental Table S6), and we appropriately discuss this data in the revised discussion (page 15).

Is it possible to grow AciCC in nude mice to test if enhancer-inhibitor strategy (such as BETi) will reduce tumor growth. This will further support the enhancer/SE hijacking hypothesis. Or, will knocking out NR4A3 block tumor growth in the nude mouse models?

> We agree that experiments in mouse xenotransplantation models would additionally strengthen the concept of enhancer hijacking and the oncogenic role of NR4A3. However, we have not identified any patient-derived AciCC cell line. The establishment of PDX mouse lines is in preparation but will not be ready for the resubmission. In addition, it was not possible to generate AciCC mouse models with stable NR4A3 knock-down/knockout as the Nr4a3 gene was not expressed endogenously in the cell lines that we isolated and immortalized from normal mice. We could not find any mouse models of AciCC which might have been used to establish primary AciCC cells. These would have been prerequisite to have to attempt stable knock-down/knockout of the Nr4a3 gene. Please see also our extensive reply to a similar suggestion made by reviewer #2.

Additional points:

- In text relating to Fig. 4, expression of human NR4A3 is suggested to increase endogenous mouse Nr4a3. However the data pertaining to Hprt and Nr4a3 expression is not available. Only human NR4A3 expression is shown, which is exogenously over-expressed.

> Former Figure 4 (now Figure 6) has been extensively revised, now demonstrating the expression of NR4A3 as well as of NR4A3 target genes Ccnd1 and Eno3 in NR4A3-transduced immortalized mouse cells both on the mRNA and protein levels. To address the expression rates of exogenously over-expressed human NR4A3 in the mouse cells we have carried out RNA-sequencing and proteomic profiling of these cells demonstrating that the levels of ectopically expressed human NR4A3 are not outrageous (Supplemental Figure 12 and Supplemental Tables 13 and 14). Indeed, mouse Nr4a3 was not expressed in the immortalized mouse salivary gland cells, this has been corrected in the text.

- NR4A3 gene body methylation being significantly lower in two of the AciCC is correlated to sustained activated expression of NR4A3. This is not clear as most previous studies have suggested that gene-body methylation positively correlated with gene activity (increased expression associated with increased gene body methylation).

> DNA methylation at gene body has indeed been observed to exhibit positive correlation with the expression level of respective genes. The seeming paradox, given the silencing

function of this mark at promoter regions, can be resolved if one considers the fact that most of the genes have alternative TSSs up- and down-stream of the canonical one. It is most likely that under normal circumstance the main purposes of gene-body methylation are to silence alternative TSS and prevent aberrant splicing. Since gene expression level of a gene measured in bulk tissue samples is impacted by the relative fraction of cells expressing this gene, its positive correlation with gene-body methylation can be readily explained through cellular heterogeneity. An alternative explanation for these observations is that aberrant transcription might be induced from several alternative gene-body TSSs. Verifying this hypothesis would require dedicated experiments, involving e.g. CAGE-Seq, which, however, we feel to be out of the scope of the present study.

Reviewer #2 (Remarks to the Author):

This manuscript describes a genomic, transcriptomic, and epigenomic profiling of salivary acinic cell carcinomas (AciCC). WGS of 6 AciCCs revealed t(4;9)(q13;q31) translocations in all cases. There were no recurrent copy number alterations or mutations. The 4q13 breakpoints clustered within the secretory Ca-binding phosphoprotein (SCPP) gene cluster and the 9q31 breakpoints were located upstream of the orphan nuclear receptor gene NR4A3. RNA seq did not reveal any functionally fusion transcripts but showed significant upregulation of NR4A3 compared to normal parotid gland (NPG). No other genes up- or downstream of NR4A3 were overexpressed compared to NPG. The 4q13 breakpoints mapped to regions with enrichment of active chromatin marks associated with abundantly expressed salivary gland genes. Moreover, active chromatin marks from the SCPP cluster were located directly at the 9q31 breakpoint region upstream of NR4A3 in AciCCs. DNA methylation throughout the NR4A3 gene was significantly lower in two out of three AciCCs compared to normal salivary gland, consistent with activation of the gene. Mouse salivary gland cells stably transduced with a wildtype NR4A3 cDNA showed increased viability and cell index compared to control cells, suggesting that NR4A3 has transforming potential in mouse cells. The authors conclude that the t(4;9) link active enhancer elements from highly expressed salivary genes in the SCPP cluster to the NR4A3 upstream region, resulting in activation of NR4A3 expression. It is also suggested that NR4A3 is an oncogenic driver in AciCC.

This is an ambitious study employing state-of-the-art methodology. The epigenetic analyses are comprehensive and solid.

> We thank the reviewer for his/her positive comment, and appreciate his/her favorable evaluation of our work.

The study is, however, mainly descriptive but contains some new information. NR4A3 is a known oncogene previously shown to be activated by chromosome translocations in sarcomas. A major concern is the lack of convincing functional evidence proving that NR4A3 is an oncogenic driver in AciCC. The authors only present physical evidence of a t(4;9) by WGS in 6 cases and show that the translocation leads to overexpression of NR4A3. No breakpoints were identified in any genes in 4q13 or 9q31 and no functional fusion transcript were identified. To substantiate the conclusion that NR4A3 is an oncogenic driver in AciCC

the authors should knock-down the expression of the gene in human AciCC cells and study the effects of knock-down on proliferation, apoptosis etc.

> We agree that the suggested NR4A3 knock-down experiments in AciCC cell line models would further establish the oncogenic role of NR4A3 in AciCC. However, no commercially available or published human patient-derived AciCC cell lines exist that could be employed in a NR4A3 knock-down experiment. A likely explanation for this is the fact that the majority of AciCCs are slowly proliferating low-grade tumors that also do not grow in nude mice or as spheroids. In contrast, high-grade transformed and fast proliferating AciCCs that could potentially be employed to generate AciCC mouse models for further stable NR4A3 knock-down are relatively rare, e.g. they represent only 5% among all AciCCs (Scherl et al. Outcomes and prognostic factors for parotid acinic cell Carcinoma: A National Cancer Database study of 2362 cases. *Oral Oncol.* 2018 Jul;82:53-60. doi: 10.1016). In regard of other groups potentially working on the molecular background of AciCC, we agree with reviewer #3 suggesting that our results should be published soon. Taken together, these obstacles made it impossible for us to perform the suggested NR4A3 knock-down experiment within an adequate time period. However, we suggest that the new data from the mouse salivary gland cells as well as a human mammary cell model both having been stably transduced to overexpress NR4A3 (see remarks to the next comment) presented in the revised manuscript should provide convincing arguments to establish the oncogenic role of NR4A3 in AciCC.

They should also study the transforming activities of NR4A3 in human salivary or mammary cells by analyzing the effects on proliferation, viability, spherogenesis, growth in soft agar etc.

> No commercially available human non-neoplastic human salivary gland cell lines exist, and the few published salivary gland cell lines either do not show the acinar (serous) differentiated phenotype, have been shown to be HeLa contaminated (e.g. HSG cell line, see Lin LC et al. *Oral Dis.* 2018 Jun 20. doi: 10.1111/odi.12920), or were not available for us upon request. Following the advice of the reviewer, in the revised manuscript we have thus extended the data from our previously presented NR4A3 transduced immortalized mouse salivary gland cell line and provide additional data from a newly generated NR4A3

transduced human mammary cell model (MCF10A cells). Both cell line models have now been tested in proliferation and cell cycle (nuclei count as well as BrdU/7AAD cell cycle) assays and consistently showed enhanced cell proliferation as well as an elevated fraction of cells in S-phase compared to non-expressing control cells (Novel Figure 6e,f). In the light of the slow growth of low-grade AciCC in patients we believe that the highly recurrent translocation and activation of NR4A3 is indeed a very, if not the earliest event in the genesis of AciCC, however, that secondary events need to occur to trigger full oncogenic transformation of affected cells. We thus believe that the proliferation data we show in the revised manuscript should be highly confirmatory of the oncogenic nature of NR4A3 and the translocation we have newly identified in AciCC. This novel data demonstrates the transforming activities of NR4A3, and is critically discussed in the revised manuscript (page 14).

Another weakness of the paper is that only one target gene of NR4A3 is studied (CCND1). Additional target genes should be analyzed in order to further characterize the molecular consequences of activation of NR4A3 in AciCC.

> We already showed a significant correlation between higher NR4A3 ChIP-seq peaks and upregulated genes in the AciCC tumors and a significant enrichment of the main NR4A3 binding motif NBRE among differential H3K27ac peaks in our manuscript, which is a strong indicator of the molecular consequence of NR4A3 activation in AciCC.

To identify further NR4A3 target genes that could be validated for the revision, we employed several approaches. First, literature research revealed only four genes that had been functionally validated as direct NR4A3 target genes in-vitro through ChIP-seq analysis and/or EMSA: These include **CCND1** (Nomiya T, Zhao Y, Gizard F, Findeisen HM, Heywood EB, Jones KL, Conneely OM, Bruemmer D. Deficiency of the NR4A neuron-derived orphan receptor-1 attenuates neointima formation after vascular injury. *Circulation*. 2009 Feb 3;119(4):577-86.), **ENO3** (Kim AY, Lim B, Choi J, Kim J. The TFG-TEC oncoprotein induces transcriptional activation of the human β -enolase gene via chromatin modification of the promoter region. *Mol Carcinog*. 2016 Oct;55(10):1411-23), **VTN** (Martí-Pàmies I, Cañes L, Alonso J, Rodríguez C, Martínez-González J. The nuclear receptor NOR-1/NR4A3 regulates the multifunctional glycoprotein vitronectin in human vascular smooth muscle cells. *FASEB J*.

2017 Oct;31(10):4588-4599.), and **PPARG** (Filion C, Motoi T, Olshen AB, Laé M, Emmett RJ, Gutmann DH, Perry A, Ladanyi M, Labelle Y. The EWSR1/NR4A3 fusion protein of extraskeletal myxoid chondrosarcoma activates the PPARG nuclear receptor gene. J Pathol. 2009 Jan;217(1):83-93).

Next, we compared published gene expression studies employing NR4A3-transduced cell systems (vascular cells, atherosclerosis, muscle cells and liver cancer) and comparing extraskeletal myxoid chondrosarcoma with *EWSR1-NR4A3* gene fusion. Apart from *CCND1* being upregulated in *NR4A3*-transduced vascular cells and hepatocellular carcinoma, we found no further gene recurrently associated with NR4A3 in the different studies (For a direct comparison of three different studies in EMC finding no overlapping gene present in all three studies, see Filion C, Labelle Y. Identification of genes regulated by the EWS/NR4A3 fusion protein in extraskeletal myxoid chondrosarcoma. Tumour Biol. 2012 Oct;33(5):1599-605.). We suggest that apart from the different methodologies, the different cellular contexts likely contribute to the marked difference in published putative NR4A3 target genes. In summary, only few functionally validated NR4A3 target genes including *CCND1*, *ENO3*, *VTN* and *PPARG* have been published, of which *VTN* encodes for a secreted protein which is difficult to analyze in a cell monoculture. We thus chose *CCND1* and *ENO3* to be further validated as NR4A3 target genes.

Regarding our own data, *ENO3* and *VTN* were within our RNA-seq list of significantly upregulated genes in the human AciCCs (Supplemental Table S7, Supplementary Figure S10) with *CCND1* being higher expressed as well but not reaching significance level. We now also show a significant upregulation of *CCND1* and *ENO3* in human AciCC tissue samples by immunohistochemistry (Supplemental Figure S11, Supplemental Table S4). Employing our in-vitro systems, we generated RNA-seq and mass spec data from our NR4A3-transduced immortalized mouse salivary gland model and control cells having been stably transduced with red firefly luciferase. *Ccnd1* and *Eno3* were upregulated on mRNA and protein level, this is now included in the revised manuscript (Figure 6, Supplementary Tables S13 and 14, Supplementary Figure S12).

In summary, combining intensive literature research, published gene expression data sets and newly data we have collected in the salivary gland cell context, we suggest that target gene specificity of NR4A3 is cell context dependent, with *CCND1/Ccnd1* and *ENO3/Eno3*

being reproducibly identified as NR4A3 transcriptional target genes also by our own data. The validation of *ENO3* as an additional NR4A3 target gene is included in the revised manuscript while the impact of NR4A3 on the expression of *CCND1* had been described already in our original submission.

The SCPP gene locus (see Introduction lines 50-51 and elsewhere) is not a specific locus (gene) but a cluster of several different genes. It should therefore not be designated the "SCPP gene locus". In addition, information about the genes included in this cluster, their functions and role in salivary glands should be added.

> We agree that the term SCPP gene cluster is a more appropriate designation for this genomic region, and this has been changed in the revised manuscript. The description of the function of these genes has been included in the introduction of the revised manuscript (first paragraph of the introduction, page 4).

The FISH image shown in Fig. 1D is difficult to interpret since the boundaries of individual nuclei are impossible to discern and therefore it is not possible to count the signals in these nuclei. The image should also include several nuclei showing the same FISH pattern.

> The old FISH image has been removed from Figure 1, and a new FISH image showing several nuclei with translocation signals has been included as supplemental figure in the revised manuscript (Supplemental Figure S2).

The quality of the H&E sections of normal parotid gland and AciCC (Fig. 4A) could be substantially improved.

> These images have been removed from Figure 4 (now Figure 6), and novel images of NR4A3, *CCND1* and *ENO3* immunostainings have been included in the revised manuscript (Supplemental Figure S11).

The authors should include more detailed information about the histology of their AciCC cases, that is whether they have a conventional histology or are cases with high-grade transformation. AciCC5 in Fig 1B looks heavily rearranged and could represent a case with high-grade transformation?

> We thank the reviewer for this very relevant suggestion. AciCC5 indeed represents a case with high-grade transformation. Additional data including histology and clinical follow-up from an extended cohort has been included in the revised manuscript (Supplemental Table S4).

Fig. 2, showing the translocation breakpoints (TXs), is somewhat difficult to understand and needs to be explained or better illustrated. Do AciCC3-6 have inversions at the chromosome 4 bp (cf. Fig 3D)? Please provide a more detailed explanation of secondary events (e.g. inversion of the SCPP gene cluster) occurring at the breakpoint sites on 4q and 9q.

> In the revised figure legend for Figure 2, the graphical visualization of the translocation breakpoints has been explained more in detail, and structural rearrangements at the breakpoint sites are included in the revised Figure 2. Presenting 9 additional AciCCs with [t(4;9)(q13;q31)] aberration identified by hybrid capture NGS in the revised manuscript, we now provide evidence for three subgroups differing by the 4q13 breakpoint location and also by the orientation (e.g. inversion) of the 4q13 genomic region. This is explained in detail in the revised manuscript (Figure 2, Figure 3, page 7-8).

Fig 2, the last panel showing the gene body DNA methylation pattern. AciCC1 looks very similar to normal parotid gland and different from AciCC 2 and 3 (cf. Fig 4B where this case clusters with normal parotid gland). Does this case contain a large amount of normal parotid gland tissue and only limited amounts of tumor tissue?

> For all samples, frozen tissue sections were evaluated for regions with tumor cell content >80% by an experienced head and neck pathologist, and only these regions were micro-dissected for DNA and RNA extraction. AciCC1 is a small low-grade conventional AciCC from an adolescent with no recurrence, and showed a highly stable genome with almost no gains or losses, and only 7 non-synonymous SNVs. Since we cannot provide another explanation,

we suggest that this case represents a very early stage AciCC with no further secondary genetic events apart from the t(4;9)(q13;q31) translocation, thus closely resembling normal parotid gland tissue in the DNA methylation and RNA expression analysis.

It would be interesting to see the expression level of NR4A3 for each AciCC and to compare it to the type of translocation (where the breakpoints are located) and to the amount of activating chromatin marks in the vicinity of NR4A3.

> The expression levels of NR4A3 (FPKM) are AciCC1: 49.78, AciCC2: 85.97, AciCC3: 192.87. We fully agree with the reviewer that the respective breakpoint region might affect expression levels of NR4A3, however, we think that the number of three samples is too low to draw further conclusions, and thus suggest not to include these values in the figure.

I suggest that the authors use at least two software programs for detection of gene fusions.

> We employed deFuse (McPherson, A. et al. deFuse: An Algorithm for Gene Fusion Discovery in Tumor RNA-Seq Data. PLoS Comput. Biol. 7, e1001138 (2011).) and additionally for the revision the tool arriba (<https://github.com/suhrig/arriba>) to detect gene fusions, and both software programs did not detect any in-frame gene fusion in the AciCC samples. This has been included in the revised methods part.

I also suggest that the authors show cell counts over time instead of cell index and viability (Fig 4 I and G). There is a big difference between the effects presented in Fig 4G and 4I.

> Following the suggestion of the reviewer we have performed two different types of proliferation assay using the stably transfected mouse salivary gland as well as human mammary MCF10A cells, now included in the extensively revised former Figure 4 (now Figure 6). First, we did microscopic count of nuclei at time points 0 and 24 hours and then computed the level differences in the counts within control and NR4A3-transfected cells. Secondly, we incubated control and NR4A3 transfected cells with BrdU and 7AAD and performed FACS analysis after cell fixation. While the former assay informed on differences in absolute numbers of cells, the latter provided information on potential alterations in

ratios of cells that were 'caught' in the different cell cycle phases. Indeed, data from both, the nuclei count and cell cycle assays, strongly support our claim that NR4A3 contributes to the enhancement of oncogenic cell growth. The data is presented in the revised manuscript in the extended Figure 6 for both cell line models.

The authors should also include a western blot of the transfected mouse cells and not only show the mRNA expression (Fig. 4H).

> In response to the reviewer's comment on validation of protein expression we tested three commercial antibodies that are advertised to detect the NR4A3 protein (Santa Cruz order numbers SC393902 and SC393903, and Biorbyt orb256728). However, even though these antibodies are advertised to specifically detect a protein with an apparent molecular weight of about 68 kDa, they did not show specific bands in the Western blots we performed with the mouse and MCF-10A cell lines used in our study. Given that we could confirm expression of NR4A3 protein by mass spectrometry we conclude that none of these antibodies is of sufficient quality. Hence, we base our claims on ectopic expression of NR4A3 in the mouse salivary gland cell line model on the results of the mass spectrometry experiment and these are unequivocal: the cell line having been stably transfected with NR4A3 does express this at the RNA (RNA-seq and qRT-PCR data) and protein (mass spectrometry) levels, while the control cell line does not.

We thus suggest to present the novel mass spectrometry data which is now included in the revised Figure 6 as well as Supplemental Table S14 and Supplemental Figure S12. Making use of the same proteomic data sets, we could identify mouse Cyclin D1 and Eno3 proteins and their higher levels in NR4A3 vs. red firefly luciferase transduced mouse salivary gland cells, with higher protein level of Cyclin D1 also shown in a Western blot (Supplemental Figure S12d).

Reviewer #3 (Remarks to the Author):

The authors describe recurring rearrangements in acinic cell carcinoma (AciCC), and perform genome, transcriptome, and epigenomic profiling, implicating highly recurrent inter-chromosomal translocations in enhancer hijacking activating the oncogenic transcription factor NR4A3. Further gene expression and in-vitro analyses are presented corroborating a likely oncogenic role of NR4A3 in AciCC via Cyclin D1, including 2.5-fold upregulation of mouse *Ccnd1* mRNA, increased Cyclin D1 protein levels and increased cell index. This is an interesting contribution describing interesting and novel finding in a rare tumour entity. It is indeed likely that the active enhancers juxtaposed to NR4A3 can drive NR4A3 gene expression. This is a timely result that should be published soon.

> We thank the reviewer for his/her positive comment, and appreciate his/her favorable evaluation of our work.

My only major criticism pertains to the wording used by the authors. The authors state that ‘the rearrangements translocate active enhancer regions from highly expressed salivary gland genes to the NR4A3 upstream region, resulting in upregulated expression and nuclear accumulation of NR4A3.’ The word “resulting” is an overstatement, and should be toned down, since the authors did not present direct experimental evidence for interaction between the enhancer elements and the NR4A3 promoter (as e.g. pursued in PMID:27869826 using 4-C sequencing). At least the remaining limitation that this test for direct interaction has not been pursued should be made clear in the Discussion.

> We agree with this comment. This limitation has now been made clear in the revised discussion (page 14-15), and the cited statement from the abstract has been revised (abstract). To further address and clarify this issue we aimed at providing circularized chromatin conformation capture combined with next generation sequencing (4C-seq) from patient tissues AciCC 1-3 using the NR4A3 breakpoint region as view point to clarify the possible interactions. Unfortunately, the remaining little amount of frozen tumor material was consumed without yielding sufficient data. We provide novel data on enhancer activity within the SPP gene cluster instead (page 9, Figure 5, Supplemental Table S6).

Additional points:

- Figure 1: I agree NR4A3 seems to be a likely target here, but this figure needs to clarify a few aspects. Please indicate the p-value that '***' refers to. Please compute p-values for other genes, such as SEC61B, which might be significant too? What is the difference in fold change between NR4A3 and other genes. Please clarify whether there is sufficient evidence to rule out that the expression of other genes may have an additional role in AciCC.

> The explanation for the *** P-value has been included in the revised manuscript (Figure legend for Figure 1). An additional Supplemental Table S3 has been included in the revised version of the manuscript showing that only NR4A3 is significantly upregulated among these genes.

- An analysis of common (cell type invariant) TADs (topologically associating domain structures) at the affected loci would be potentially useful for the readers. I assume the breakpoints will likely affect TAD boundaries with potentially insulating effect. It should be reassured though that there are no TAD boundaries between the active enhancer elements shown in Figure 3 and the target gene NR4A3. Datasets calling TAD boundaries present across distinct cell types could be used for such analysis.

> We fully agree with this excellent comment. TAD boundaries and HiC contact maps from published data sets have been included in the revised figure 2, and show indeed a strong correlation with our novel ChIP-seq and RNA-Seq data from normal salivary gland tissue. This was also very helpful for the further identification and characterization of three subgroups according to different 4q13 breakpoint regions. This important observation has been included in the revised manuscript (page 6 and 7, Figure 2, novel Figure 5).

Description of the protocol used for circularized chromatin conformation capture combined with next generation sequencing (4C-seq) (for review process only)

All remaining frozen tumor biopsies from AciCC 1-3 (70 – 120 mg) were treated with collagenase A (Roche; catalog no. 10103578001) to obtain tumor cell suspensions as described by Matelot et al. 2016. Circularized chromatin conformation capture combined with next generation sequencing (4C-seq) was adapted from the protocol described by van de Werken et al. 2012. In brief, chromatin of suspended tumor cells was crosslinked with formaldehyde and subjected to a first restriction digestion using restriction enzyme HindIII suited for cutting in crosslinked chromatin and known to cut within the NR4A3 promoter, the 4C viewpoint. After a first ligation which favorably joined the ends of the NR4A3 promoter-derived HindIII fragment with themselves but also with unknown HindIII fragments located in close vicinity due to chromatin crosslinking and, hence, likely derived from enhancer regions, the chromatin was decrosslinked. The resulting circularized DNA molecules were subjected to a second restriction digestion using the frequent cutter DpnII which also cuts in the viewpoint and with high likelihood in the unknown connected HindIII fragments. A second ligation with diluted DpnII-cut DNA favored self-ligation and, thus, circularization of DpnII fragments. Inverse PCR using primers pointing outwards of the known NR4A3 promoter-derived HindIII-DpnII fragment were employed to generate PCR products of the joined unknown DNA fragments, which were subsequently analyzed by high-throughput next generation sequencing on a HiSeq2000 sequencer (v4, single read 50 bp).

Results

According to the low amount of input material, DNA yield after the two restriction digestion/ligation rounds was low (35 - 120 ng). For comparison, in the standard 4C protocol with 10 million cultivated cells, the DNA yield after the two restriction digestion/ligation rounds usually amounts to 20-30 µg. As a consequence of the low yield from the AciCC biopsies, the amount of library PCR products was also low with not more than 3-4 ng per PCR using each half of the available template as input. The PCR libraries were pooled, and the pool was sequenced in one lane. Bioinformatics analysis revealed that although 72,205,232 reads were generated, the vast majority mapped to PhiX (72,065,182; 99.7%) and after demultiplexing and alignment, only 28,085, 33,639 and 519 unique reads remained for

AciCC1, AciCC2 and AciCC3, respectively. Of these, the majority mapped to the NR4A3 view point or to the cloning vector pFosill, demonstrating that the assay has worked in principle, but that according to the low tissue input the amount of captured chromatin was too low to yield sufficient fragments to generate a map of chromatin interactions with the NR4A3 view point.

Primers

4C-Seq primer Sequence Position (hg19)

Hind_NR4A3GT

AATGATACGGCGACCACCGAACACTCTTCCCTACACGACGCTCTCCGATCTGTGGTTCTCT
GGAAGGAAGCTT chr9:102,585,175

HindNR4A3AC

AATGATACGGCGACCACCGAACACTCTTCCCTACACGACGCTCTCCGATCTACGGTTCTCT
GGAAGGAAGCTT chr9:102,585,175

HindNR4A3GC

AATGATACGGCGACCACCGAACACTCTTCCCTACACGACGCTCTCCGATCTGCGGTTCTCT
GGAAGGAAGCTT chr9:102,585,175

HindNR4A3CG

AATGATACGGCGACCACCGAACACTCTTCCCTACACGACGCTCTCCGATCTCGGGTTCTCT
GGAAGGAAGCTT chr9:102,585,175

DpnPrim1 CAAGCAGAAGACGGCATAACGACACAAATCCCGGTCCTCCTC chr9:102,584,943

References

Matelot M, Noordermeer D. Determination of High-Resolution 3D Chromatin Organization Using Circular Chromosome Conformation Capture (4C-seq). *Methods Mol Biol* 1480, 223-241 (2016).

van de Werken HJ, et al. Robust 4C-seq data analysis to screen for regulatory DNA interactions. *Nat Methods* 9, 969-972 (2012).

Reviewer #1 (Remarks to the Author):

The authors have addressed key issues of this reviewer. Considering a rare tumor, with limited samples, the authors have addressed the major issues with the available material. The paper seems ready for publication. Congratulations.

Reviewer #2 (Remarks to the Author):

The authors have submitted a substantially revised and improved version of the manuscript. Also the figures are now more informative. However, my main concern remains that this is a mainly descriptive study lacking convincing functional evidence to support the authors conclusion that NR4A3 is an oncogenic driver in AciCC. The last sentence in the Abstract should therefore be modified since the authors do not provide evidence showing that AciCC in fact drives the growth of AciCC. Similar statements in other parts of the paper should also be downplayed to better reflect the results obtained.

Minor comments:

On p 13, lines 261-262, the authors state that "thus confirming a transcriptional effect of NR4A3 upregulation on its known target genes in our AciCC models". This statement is incorrect since the authors do not have any validated models mimicking AciCC.

The authors should acknowledge in the Discussion that the HTN3 gene has already been identified as part of a gene fusion in AciCC (ref 9).

The t(4;9) translocation does not seem to be entirely specific for AciCC since an identical translocation or translocations with similar breakpoints have been found also in other neoplasms (see Mitelman Database of Chromosome Aberrations and Gene Fusions in Cancer, <https://cgap.nci.nih.gov/Chromosomes/Mitelman>).

Reviewer #3 (Remarks to the Author):

The authors also are now able to show that rearranged elements qualify as super enhancers, and additional experiments prove enhancer activity through a luciferase reporter assay. Evidence for driver activity of NR4A3 was additionally shown in cellular systems. Although owing to the lack of sample material 4C-Seq could not be performed, I am satisfied with the evidence presented for enhancer hijacking and recommend publication of this paper. Clearly 4C-Seq is a technique with quite “extraordinary” sample requirements (amount of cells needed), and in the absence of suitable model systems this technique is really not very well suited to study disease samples (in the light of that, I am satisfied with what the authors have done here).

Reviewers' comments:

Reviewer #1 (Remarks to the Author):

The authors have addressed key issues of this reviewer. Considering a rare tumor, with limited samples, the authors have addressed the major issues with the available material. The paper seems ready for publication. Congratulations.

> We again thank the reviewer for his/her very positive comment and for his/her very valuable help during the revision process.

Reviewer #2 (Remarks to the Author):

The authors have submitted a substantially revised and improved version of the manuscript. Also the figures are now more informative. However, my main concern remains that this is a mainly descriptive study lacking convincing functional evidence to support the authors conclusion that NR4A3 is an oncogenic driver in AciCC. The last sentence in the Abstract should therefore be modified since the authors do not provide evidence showing that AciCC in fact drives the growth of AciCC. Similar statements in other parts of the paper should also be downplayed to better reflect the results obtained.

> We again thank the reviewer for his/her positive comment and for his/her very valuable help during the revision process. The last sentence in the abstract and similar statements have been modified accordingly to reflect the results from the study (page 3 lines 44-49; page 11 line 223; page 14 lines 296-299; page 15 line 309).

Minor comments:

On p 13, lines 261-262, the authors state that "thus confirming a transcriptional effect of NR4A3 upregulation on its known target genes in our AciCC models". This statement is incorrect since the authors do not have any validated models mimicking AciCC.

> This statement has been modified (now page 14 lines 296-299).

The authors should acknowledge in the Discussion that the HTN3 gene has already been identified as part of a gene fusion in AciCC (ref 9).

> This is now acknowledged in the discussion (page 15, lines 322-325).

The t(4;9) translocation does not seem to be entirely specific for AciCC since an identical translocation or translocations with similar breakpoints have been found also in other neoplasms (see Mitelman Database of Chromosome Aberrations and Gene Fusions in Cancer, <https://cgap.nci.nih.gov/Chromosomes/Mitelman>).

> We applied a search for the [t(4;9)(q13;q31)] rearrangement using the “Mitelman Cases Quick Searcher” with the search term “*t(4;9)(q13;q31)” and found only a single case with this specific rearrangement among the 68,598 cases in this database (see screenshot of results below). Using the search term “*t(4;9)” revealed 125 cases, mostly lymphoma and leukaemia cases harboring a t(4;9) rearrangement involving other chromosomal regions, e.g. t(4;9)(p14;q12). Since we show in our current study that in AciCCs the genomic breakpoints cluster among only 340 kb at chromosomal region 4q13 and among <300 kb at chromosomal region 9q31, we suggest that a search for tumor samples with t(4;9) rearrangement is not specific enough to warrant a comparison with the highly recurrent and focused [t(4;9)(q13;q31)] rearrangement in AciCCs.

Sreekantaiah et al 1991, Cancer Res

Case No. 36	Lipoma	Soft tissue
	46,XX,t(4;9)(q13;q31)	

Reviewer #3 (Remarks to the Author):

The authors also are now able to show that rearranged elements qualify as super enhancers, and additional experiments prove enhancer activity through a luciferase reporter assay. Evidence for driver activity of NR4A3 was additionally shown in cellular systems. Although owing to the lack of sample material 4C-Seq could not be performed, I am satisfied with the evidence presented for enhancer hijacking and recommend publication of this paper. Clearly 4C-Seq is a technique with quite “extraordinary” sample requirements (amount of cells needed), and in the absence of suitable model systems this technique is really not very well suited to study disease samples (in the light of that, I am satisfied with what the authors have done here).

> We again thank the reviewer for his/her very positive comment and for his/her very valuable help during the revision process.